# Kronecker-Factored Approximate Curvature for Modern Neural Network Architectures

**Runa Eschenhagen**[*,1]     **Alexander Immer**[2,3]     **Richard E. Turner**[1]
**Frank Schneider**[4,5]     **Philipp Hennig**[4,5]

[1]University of Cambridge
[2]Department of Computer Science, ETH Zurich
[3]Max Planck Institute for Intelligent Systems
[4]University of Tübingen
[5]Tübingen AI Center

## Abstract

The core components of many modern neural network architectures, such as transformers, convolutional, or graph neural networks, can be expressed as linear layers with *weight-sharing*. Kronecker-Factored Approximate Curvature (K-FAC), a second-order optimisation method, has shown promise to speed up neural network training and thereby reduce computational costs. However, there is currently no framework to apply it to generic architectures, specifically ones with linear weight-sharing layers. In this work, we identify two different settings of linear weight-sharing layers which motivate two flavours of K-FAC – *expand* and *reduce*. We show that they are exact for deep linear networks with weight-sharing in their respective setting. Notably, K-FAC-reduce is generally faster than K-FAC-expand, which we leverage to speed up automatic hyperparameter selection via optimising the marginal likelihood for a Wide ResNet. Finally, we observe little difference between these two K-FAC variations when using them to train both a graph neural network and a vision transformer. However, both variations are able to reach a fixed validation metric target in 50-75% of the number of steps of a first-order reference run, which translates into a comparable improvement in wall-clock time. This highlights the potential of applying K-FAC to modern neural network architectures.

## 1 Introduction

One of the key driving forces behind the success of deep learning is arguably the development and scaling of novel deep neural network (DNN) architectures like transformers (Vaswani et al., 2017) or graph neural networks (Battaglia et al., 2018; Scarselli et al., 2009). While the landscape of neural network architectures seems vast, their core building blocks like attention, graph, recurrent, and convolutional layers can be expressed as simple linear layers with *weight-sharing*. For example, the weight-sharing can happen over a sequence of tokens, e.g. in language modelling, over image patches, or by tying weights in the forward pass. The perspective of expressing the core neural network operations as linear operations has been formalised in a different context with the Tensor Programs framework (Yang, 2019), which develops a language to express arbitrary neural network computations as a composition of simple matrix multiplications and coordinate-wise nonlinearities. One downside of relying on increasing the scale of the models to achieve better performance is the increased compute cost. To decrease these costs, we can try to improve the efficiency of the optimisation algorithms, which motivates exploring second-order methods like Newton's method and natural gradient descent (Amari, 1998), which uses the Fisher information matrix, or short, the Fisher.

---

[*]Correspondence to: re393@cam.ac.uk.

37th Conference on Neural Information Processing Systems (NeurIPS 2023).

Kronecker-Factored Approximate Curvature (Heskes, 2000; Martens & Grosse, 2015, K-FAC) is an approximation to the Fisher, or for common loss functions equivalently, the generalised Gauss-Newton matrix (GGN). It has first been derived in the context of optimisation for linear layers and later for convolutional (Grosse & Martens, 2016) and recurrent (Martens et al., 2018) layers. In deep learning, using the Fisher/GGN instead of the Hessian has been the de-facto standard; in turn, K-FAC has arguably been one of the most popular approximations of the Fisher/GGN, probably due to its relative efficiency as it approximates each layer's Fisher/GGN independently with a Kronecker product. While K-FAC has also been used for transformers (Grosse et al., 2023; Osawa et al., 2022; Pauloski et al., 2021; Zhang et al., 2019a) and a graph neural network (Izadi et al., 2020), there is no theoretical framework for these cases and for applying K-FAC to new architectures.

**Contributions.** We propose such a framework by leveraging the idea of linear layers with weight-sharing. This concept reveals the additional structure in the Fisher/GGN due to the weight-sharing that has to be explicitly considered when applying K-FAC. We identify two settings of such layers that motivate two different flavours of the K-FAC approximation: *K-FAC-expand* and *K-FAC-reduce*. The former corresponds to the Kronecker Factors for Convolution (KFC) approximation in Grosse & Martens (2016), to the simplest approximation for recurrent layers proposed in Martens et al. (2018), and has been used for transformers (Grosse et al., 2023; Osawa et al., 2022; Pauloski et al., 2021; Zhang et al., 2019a) – though without discussion or motivation. Notably, K-FAC-reduce is exact for certain settings, like deep linear networks with convolutions and average pooling, under the same conditions that K-FAC is exact for deep linear networks without weight-sharing, whereas the currently used K-FAC-expand is not. In practice, both approximations can be used in each setting and K-FAC-reduce generally has a lower computational and memory complexity than K-FAC-expand. We empirically verify this speed difference with a Wide ResNet on CIFAR-10. Moreover, we show that the two K-FAC variations applied to a graph neural network and vision transformer can reach a fixed validation metric target in 50-75% of the steps of a first-order reference method, which translates into an almost equivalent decrease in wall-clock time. Although this does not demonstrate that K-FAC is a superior training algorithm, it indicates the potential of extending K-FAC to modern deep learning architectures. In general, K-FAC can be used as a drop-in Hessian approximation, e.g., in Laplace approximations for Bayesian deep learning (Daxberger et al., 2021; MacKay, 1992; Ritter et al., 2018) or in natural gradient variational inference (Khan et al., 2018; Zhang et al., 2018). To demonstrate this, we show that K-FAC-reduce can speed up automatic weight decay selection via approximate marginal likelihood optimisation, compared to the previously used K-FAC-expand.

## 2 Background

We consider the supervised learning setup with a dataset $\mathcal{D}$ of $N$ i.i.d. samples $\{\boldsymbol{x}_n, \boldsymbol{y}_n\}_{n=1}^N$, with $\boldsymbol{x}_n \in \mathbb{R}^D$ and $\boldsymbol{y}_n \in \mathbb{R}^C$, a neural net $f_{\boldsymbol{\theta}} : \mathbb{R}^D \to \mathbb{R}^C$, parameterised with $\boldsymbol{\theta} \in \mathbb{R}^P$, and a loss function $\ell : \mathbb{R}^C \times \mathbb{R}^C \to \mathbb{R}$, which is often equivalent to a negative log likelihood, i.e. $\ell(\boldsymbol{y}, f_{\boldsymbol{\theta}}(\boldsymbol{x})) = -\log p(\boldsymbol{y}|f_{\boldsymbol{\theta}}(\boldsymbol{x}))$. The DNN is usually trained with a (stochastic) gradient-based iterative update rule, in its simplest form $\boldsymbol{\theta}_{t+1} = \boldsymbol{\theta}_t - \alpha \nabla_{\boldsymbol{\theta}_t} \ell(\boldsymbol{y}, f_{\boldsymbol{\theta}_t}(\boldsymbol{x}))$, where $t$ indicates the iteration and $\alpha$ is the learning rate. If we assume a local quadratic approximation of the loss around the current parameter iterate $\boldsymbol{\theta}_t$, we get a *preconditioned* gradient update step, i.e.

$$\boldsymbol{\theta}_{t+1} = \boldsymbol{\theta}_t - \alpha \, \boldsymbol{C}_t^{-1} \nabla_{\boldsymbol{\theta}_t} \ell(\boldsymbol{y}, f_{\boldsymbol{\theta}_t}(\boldsymbol{x})), \tag{1}$$

where $\boldsymbol{C}_t \in \mathbb{R}^{P \times P}$ is the symmetric positive definite (p.d.) preconditioner. Setting $\boldsymbol{C}_t$ to the Hessian of the loss w.r.t. the model parameters $\boldsymbol{H}_t := \nabla^2_{\boldsymbol{\theta}_t} \ell(\boldsymbol{y}, f_{\boldsymbol{\theta}_t}(\boldsymbol{x}))$ yields a simplified version of the classic Newton's method; however, since $\boldsymbol{H}_t$ is expensive to compute, store, and not guaranteed to be p.d., it is not commonly used in deep learning.

### 2.1 Second-order optimisation in deep learning

The term "second-order" is used ambiguously in the literature: it can refer to using second-order derivatives like the Hessian of the loss w.r.t. the parameters of the model, or approximations thereof, like the GGN, which however only contains second-order derivatives of the loss w.r.t. the model outputs and not the parameters. It can also refer to using some variation of the second moment of the (average) mini-batch gradient, which blurs the line between commonly used methods that are usually considered "first-order" methods like Adam (Kingma & Ba, 2015).

There have been numerous works on "second-order" methods in deep learning. For example, Martens (2010) proposes to use Hessian-free optimisation, Gupta et al. (2018) introduce an architecture-agnostic structured version of AdaGrad (Duchi et al., 2011) called Shampoo, Goldfarb et al. (2020) develop K-BFGS, a Kronecker-factored stochastic quasi-Newton method, and Ren & Goldfarb (2021) introduce Tensor Normal Training, which is similar to Shampoo, but approximates the Fisher; just to name a few among many more methods (Yang et al., 2022, 2023; Yao et al., 2021). While these methods have been shown to require fewer steps than commonly used first-order methods to reach similar performance in many settings, the mechanism behind their benefits is not fully understood. It could be orthogonal to unique properties of the true GGN or Fisher but might be more related to being able to adapt to the stochastic gradient noise (Kunstner et al., 2019) or to the fact that K-FAC with a specific but commonly used form of damping can be viewed as an approximation to gradient descent on neurons instead of weights (Benzing, 2022). In any case, generalising K-FAC to modern architectures is a promising direction because it is used in many training algorithms that can potentially speed up training and train a wider range of architectures. Moreover, it can be used to study the mechanisms behind the benefits of second-order methods in deep learning.

## 2.2 Linear weight-sharing layers

Here, we present three examples of how the core building blocks of modern DNN architectures can be expressed as linear layers with weight-sharing. DNNs have a layered structure, i.e. they can be written as $f_{\boldsymbol{\theta}} = f_{\boldsymbol{\theta}_L} \circ \ldots \circ f_{\boldsymbol{\theta}_\ell} \circ \ldots \circ f_{\boldsymbol{\theta}_1}$, with $\boldsymbol{\theta} = \text{concat}(\boldsymbol{\theta}_1, \ldots, \boldsymbol{\theta}_\ell, \ldots, \boldsymbol{\theta}_L)$ and $L$ the number of layers; $\text{concat}(\cdot, \ldots, \cdot)$ concatenates vector inputs to a larger vector. For a linear layer, we have $f_{\boldsymbol{\theta}_\ell}(\boldsymbol{x}) = \phi(\boldsymbol{W}_\ell \boldsymbol{x} + \boldsymbol{b}_\ell)$, where $\boldsymbol{x} \in \mathbb{R}^{P_{\ell,\text{in}}}$, $\boldsymbol{W}_\ell \in \mathbb{R}^{P_{\ell,\text{out}} \times P_{\ell,\text{in}}}$, $\boldsymbol{b} \in \mathbb{R}^{P_{\ell,\text{out}}}$, $\boldsymbol{\theta}_\ell = \text{concat}(\text{vec}(\boldsymbol{W}_\ell), \boldsymbol{b}_\ell) \in \mathbb{R}^{P_\ell}$, and $P_\ell = P_{\ell,\text{out}} P_{\ell,\text{in}} + P_{\ell,\text{out}}$; $\text{vec}(\cdot)$ vectorises a matrix by concatenating its column vectors and $\phi$ is an element-wise nonlinearity. For simplicity, we subsume the bias into the weights. We are interested in linear layers with weights shared across an additional input dimension of size $R$, i.e. with inputs $\boldsymbol{X} \in \mathbb{R}^{R \times D}$.[1] The linear layer is applied to this input as $\boldsymbol{X}\boldsymbol{W}^\top$ and the weight matrix $\boldsymbol{W}$ is shared across the additional first dimension, i.e. every element in $\boldsymbol{W}$ contributes to each row of the output.

**Example 1: Attention.** The transformer architecture is popular in vision (Dosovitskiy et al., 2021), language (Brown et al., 2020), and spatio-temporal modelling (Bi et al., 2022). The inputs and outputs of the attention operation (Bahdanau et al., 2015) in transformers are exactly shaped as described above, where $R$ is the sequence length, e.g. the number of tokens in language data or the number of image patches in vision. For the $i$-th row of the output matrix $\boldsymbol{O}$ and the $r$-th row of the inputs $\boldsymbol{X}$, the attention operation is generally defined as

$$\boldsymbol{o}_i = \sum_{r=1}^{R} A_{i,r} \boldsymbol{x}_r, \tag{2}$$

where $\boldsymbol{A} \in \mathbb{R}^{R \times R}$ is the *attention matrix*, which is normalised such that $\sum_{r=1}^{R} A_{i,r} = 1$; alternatively, we can write it using matrix operations as $\boldsymbol{O} = \boldsymbol{A}\boldsymbol{X}$. Intuitively, the outputs are a weighted sum of the inputs, where each weighting factor indicates the importance of the $r$-th sequence element to the element with index $i$. The most common choice for $\boldsymbol{A}$ is a variation of so-called self-attention, defined as $\boldsymbol{A} = \text{softmax}_{\text{row}}(\boldsymbol{X}\boldsymbol{W}_Q^\top \boldsymbol{W}_K \boldsymbol{X}^\top)$, where $\boldsymbol{W}_Q \in \mathbb{R}^{P_{\text{out}} \times D}$ and $\boldsymbol{W}_K \in \mathbb{R}^{P_{\text{out}} \times D}$ are weight matrices and $\text{softmax}_{\text{row}}(\cdot)$ applies the softmax function $\text{softmax}(\boldsymbol{x})_i := \exp(x_i) / \sum_{j=1}^{J} \exp(x_j)$ for an input $\boldsymbol{x} \in \mathbb{R}^J$ to each row of the input matrix. This corresponds to a dot-product similarity measure of two independent linear projections of the input, allowing for asymmetric relationships between sequence elements. The two weight matrices $\boldsymbol{W}_Q, \boldsymbol{W}_K$, and any linear projection of the attention output are applied to all $R$ sequence elements – they are linear weight-sharing layers.

**Example 2: Convolution.** A typical input to a 2d convolution is an image $\boldsymbol{X}^I \in \mathbb{R}^{C_{\text{in}} \times H \times W}$, where $C_{\text{in}}$ is the number of input channels, $H$ is the height, and $W$ the width of the image. We choose a kernel of size $K \times K$, such that we have a rank-4 weight tensor of shape $C_{\text{out}} \times C_{\text{in}} \times K \times K$, where $C_{\text{out}}$ is the number of output channels. For simplicity, following Grosse & Martens (2016), we assume that the convolution is performed with a stride of 1 and that the padding is chosen such that the height and width of the inputs are maintained, i.e. the output image will be $\boldsymbol{O}^I \in \mathbb{R}^{C_{\text{out}} \times H \times W}$.

---

[1]Our framework also applies to weight-sharing where the weights are used multiple times during the forward pass; however, we focus the presentation on an input with an explicit weight-sharing dimension.

The elements of the output tensor of the convolution are then defined as

$$o_{d,i,j}^I = \sum_{c=1}^{C_{\text{in}}} \sum_{k=1}^{K} \sum_{k'=1}^{K} X_{c,i+k-1,j+k'-1}^I W_{d,c,k,k'}; \tag{3}$$

the input elements $X_{c,i+k-1,j+k'-1}$ are defined as 0 outside of the image's boundaries. Now, we can *unfold* the input tensor to a matrix $\boldsymbol{X} \in \mathbb{R}^{HW \times C_{\text{in}}K^2}$, where we have extracted and vectorised the $K \times K$ sized patches for each of the $H \cdot W$ spatial locations. The weight tensor can be reshaped to the matrix $\boldsymbol{W} \in \mathbb{R}^{C_{\text{out}} \times C_{\text{in}}K^2}$. Using these quantities, we can write the convolution as $\boldsymbol{O} = \boldsymbol{X}\boldsymbol{W}^{\mathsf{T}} \in \mathbb{R}^{HW \times C_{\text{out}}}$. Again, we can recognise the canonical form of a linear weight-sharing layer – here, with $R = HW$, i.e. the weights are shared across all spatial locations.

**Example 3: Graph neural network layer.** We follow Battaglia et al. (2018) since their formulation encompasses many graph neural network (GNN) layers. For the full example and notation, please refer to Appendix A. Once again, the core operations in these kinds of models can be expressed as linear weight-sharing layers. Here, the weights are shared across all $R = N_V$ nodes or $R = N_E$ edges of a graph, depending on whether node or edge features are being updated.

## 2.3 The generalised Gauss–Newton and the Fisher matrix

To avoid computing the Hessian, approximations like the generalised Gauss–Newton matrix (GGN) are usually used in deep learning (Botev et al., 2017; Martens, 2010; Schraudolph, 2002). Defining $\boldsymbol{\Lambda}(f_{\boldsymbol{\theta}}(\boldsymbol{x}_n)) := \nabla_{f_{\boldsymbol{\theta}}}^2 \ell(\boldsymbol{y}_n, f_{\boldsymbol{\theta}}(\boldsymbol{x}_n)) \in \mathbb{R}^{C \times C}$, we have

$$\mathbf{GGN}(\boldsymbol{\theta}) = \sum_{n=1}^{N} \boldsymbol{J}_{\boldsymbol{\theta}} f_{\boldsymbol{\theta}}(\boldsymbol{x}_n)^{\mathsf{T}} \boldsymbol{\Lambda}(f_{\boldsymbol{\theta}}(\boldsymbol{x}_n)) \boldsymbol{J}_{\boldsymbol{\theta}} f_{\boldsymbol{\theta}}(\boldsymbol{x}_n), \tag{4}$$

which is equivalent to the Hessian when $f_{\boldsymbol{\theta}}$ is a linear model or the residual $g^{-1}(f_{\boldsymbol{\theta}}(\boldsymbol{x}_n)) - \boldsymbol{y}_n$ is zero for all data points, where $g^{-1}$ is the inverse link function, e.g. the softmax function for classification. For all likelihoods of the exponential family with natural parameterisation the GGN coincides with the Fisher information matrix (Martens, 2014; Wang, 2010). This is, for example, the case for the common cross-entropy loss corresponding to a categorical likelihood and the mean-square error loss corresponding to a Gaussian likelihood. Setting $\boldsymbol{C}_t$ in Equation (1) to the Fisher leads to natural gradient descent (Amari, 1998). We will only state our derivations for the GGN explicitly, but they follow analogously for the Fisher.

## 2.4 Kronecker-Factored Approximate Curvature

The GGN is a semi-p.d. approximation to the Hessian, of size $P \times P$, which is still prohibitively large for DNNs. K-FAC was proposed as an efficient approximation to a neural network's Fisher (Heskes, 2000; Martens & Grosse, 2015) and GGN (Botev et al., 2017) matrix.[2] First, we only approximate the GGN for each layer, ignoring cross-layer interactions, which results in a block-diagonal matrix. We define $\boldsymbol{J}_{\boldsymbol{\theta}_{\ell}}(\boldsymbol{x}_n) := \boldsymbol{J}_{\boldsymbol{\theta}_{\ell}} f_{\boldsymbol{\theta}}(\boldsymbol{x}_n) = \boldsymbol{J}_{\boldsymbol{s}_{\ell,n}} f_{\boldsymbol{\theta}}(\boldsymbol{x}_n) \boldsymbol{J}_{\boldsymbol{\theta}_{\ell}} \boldsymbol{s}_{\ell,n} \in \mathbb{R}^{C \times P_{\ell}}$ as the Jacobian of the model outputs w.r.t. the parameters of the $\ell$-th layer. Now we can write $\boldsymbol{s}_{\ell,n} = \boldsymbol{W}_{\ell} \boldsymbol{a}_{\ell,n} = (\boldsymbol{a}_{\ell,n}^{\mathsf{T}} \otimes \mathbf{I}_{P_{\ell,\text{out}}}) \text{vec}(\boldsymbol{W}_{\ell})$ and with this we have $\boldsymbol{J}_{\boldsymbol{\theta}_{\ell}} \boldsymbol{s}_{\ell,n} = \boldsymbol{a}_{\ell,n}^{\mathsf{T}} \otimes \mathbf{I}_{P_{\ell,\text{out}}}$. Additionally, by defining $\boldsymbol{b}_{\ell,n} := \boldsymbol{J}_{\boldsymbol{s}_{\ell,n}} f_{\boldsymbol{\theta}}(\boldsymbol{x}_n)^{\mathsf{T}} \in \mathbb{R}^{P_{\ell,\text{out}} \times C}$ as the transposed Jacobian of the model outputs w.r.t. the pre-activations of the $\ell$-th layer, we note that $\boldsymbol{J}_{\boldsymbol{\theta}_{\ell}}(\boldsymbol{x}_n)^{\mathsf{T}} = (\boldsymbol{a}_{\ell,n}^{\mathsf{T}} \otimes \mathbf{I}_{P_{\ell,\text{out}}})^{\mathsf{T}} \boldsymbol{b}_{\ell,n} = \boldsymbol{a}_{\ell,n} \otimes \boldsymbol{b}_{\ell,n}$. K-FAC approximates the GGN of the $\ell$-th layer as

$$\mathbf{GGN}(\boldsymbol{\theta}_{\ell}) \approx \underbrace{\left[\frac{1}{N} \sum_{n=1}^{N} \boldsymbol{a}_{\ell,n} \boldsymbol{a}_{\ell,n}^{\mathsf{T}}\right]}_{=:\boldsymbol{A}_{\ell}} \otimes \underbrace{\left[\sum_{n=1}^{N} \boldsymbol{b}_{\ell,n} \boldsymbol{\Lambda}(f_{\boldsymbol{\theta}}(\boldsymbol{x}_n)) \boldsymbol{b}_{\ell,n}^{\mathsf{T}}\right]}_{=:\boldsymbol{B}_{\ell}}, \tag{5}$$

where we have replaced the (transposed) Jacobians in the definition of the GGN in Equation (4) by $\boldsymbol{a}_{\ell,n} \otimes \boldsymbol{b}_{\ell,n}$ and approximated the sum of Kronecker products with a Kronecker product of sums.[3]

---

[2]In the literature, the name "K-FAC" is used to refer to the approximation of the Fisher/GNN and to a specific algorithm using this approximation (Martens & Grosse, 2015); here, we use it to refer to just the approximation.

[3]Note that the Kronecker product $\boldsymbol{A} \otimes \boldsymbol{B}$ is in reverse order compared to in Figure 1, because we concatenate the columns in the derivation and the rows in the implementation when flattening a matrix with $\text{vec}(\cdot)$.

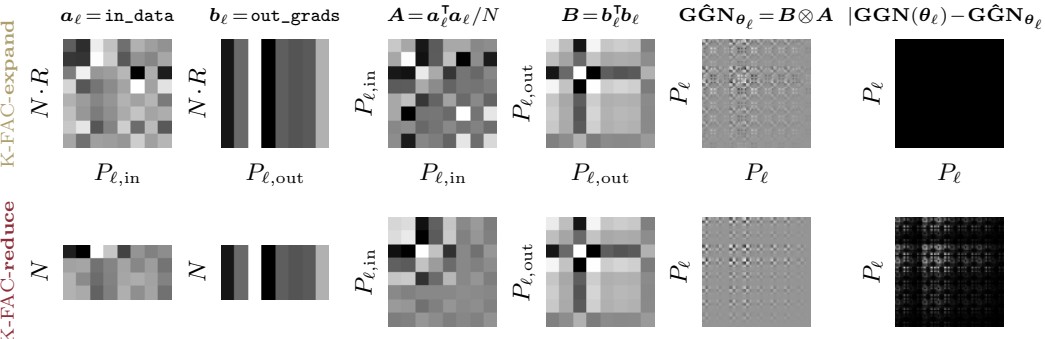

Figure 1: **Visualisation of K-FAC-expand and K-FAC-reduce in the expand setting.** The shown quantities are for a single layer within a deep linear network. We have $N = 4, R = 2, P_{\ell,\text{in}} = 8, P_{\ell,\text{out}} = 8$, and $P_\ell = P_{\ell,\text{in}} \cdot P_{\ell,\text{out}} = 64$. As we have seen in Section 3.2, K-FAC-expand is exact for the expand case in this setting and K-FAC-reduce is not. For better visibility, the color scale is not the same for all quantities, except for the approximation error (*right*) where black represents zero.

## 3   K-FAC for linear weight-sharing layers

In this section, we propose to distinguish between two different linear weight-sharing settings when applying K-FAC. The two settings motivate two corresponding approximations, *K-FAC-expand* and *K-FAC-reduce*. However, *both approximations can be applied in each setting*, see Listing 1 for an illustration with code. More details on the derivations and the proofs for Proposition 1 and Proposition 2 can be found in Appendix B.

### 3.1   The expand and reduce settings

Within a network, linear layers with weight-sharing can be classified based on the point where the weight-sharing dimension is aggregated. Specifically, there are three possible aggregation points, which will define the setting for the $\ell$-th layer:

(i) *Before* the $\ell$-th layer, i.e. $\boldsymbol{A}_{\ell,n} \in \mathbb{R}^{R \times P_{\ell,\text{in}}}$ is reduced to $\tilde{\boldsymbol{a}}_{\ell,n} \in \mathbb{R}^{P_{\ell,\text{in}}}$ before being multiplied with the weight matrix of the layer $\boldsymbol{W}_\ell$. $\rightarrow$ We are in the setting of a regular linear layer and no additional considerations are necessary when using K-FAC.

(ii) *After the per-example loss*, i.e. there will be $N \cdot R$ labels and outputs of the model. The per-example loss is applied to each of the $N \cdot R$ output-label pairs. $\rightarrow$ This is the *expand* setting, as the loss for each of the $N$ data points is expanded with $R$ terms.

(iii) *In between* the pre-activation of the $\ell$-th layer $\boldsymbol{S}_{\ell,n} \in \mathbb{R}^{R \times P_{\ell,\text{out}}}$ and the model output $f_{\boldsymbol{\theta}}(\boldsymbol{x}_n)$, i.e. before the final aggregation over the per-example losses. $\rightarrow$ This is the *reduce* setting, as all weight-sharing dimensions have been reduced during the forward pass.

If we assume a single weight-sharing dimension which will be reduced once, the number of loss terms determines which setting applies to all linear weight-sharing layers within the model. So for our purposes, *we can identify the setting we are in simply by looking at the form of the loss.*

### 3.2   K-FAC-expand

The expand setting can be characterised by a loss with $N \cdot R$ terms, similar to $N \cdot R$ i.i.d. examples, $\mathcal{L}_{\text{expand}}(f_{\boldsymbol{\theta}}, \mathcal{D}) := -\sum_{n=1}^{N} \sum_{r=1}^{R} \log p(\boldsymbol{y}_{n,r} | f_{\boldsymbol{\theta}}(\boldsymbol{x}_n)_r)$, where $f_{\boldsymbol{\theta}}(\boldsymbol{x}_n)_r$ is the $r$-th row of the model output $f_{\boldsymbol{\theta}}(\boldsymbol{x}_n) \in \mathbb{R}^{R \times C}$ and $\boldsymbol{y}_{n,r}$ is the $r$-th row of the label $\boldsymbol{Y}_n \in \mathbb{R}^{R \times C}$. A typical example of this type of loss function is language translation, where $N$ is the dataset size and $R$ is the sequence length.

We can express the Jacobian of the $r$-th row of the model output $f_{\boldsymbol{\theta}}(\boldsymbol{x}_n) \in \mathbb{R}^{R \times C}$ w.r.t. $\boldsymbol{\theta}_\ell$ as $\boldsymbol{J}_{\boldsymbol{\theta}_\ell}(\boldsymbol{x}_n)_r = \sum_{m=1}^{R} \boldsymbol{J}_{\boldsymbol{s}_{\ell,n,m}} f_{\boldsymbol{\theta}}(\boldsymbol{x}_n)_r \boldsymbol{J}_{\boldsymbol{\theta}_\ell} \boldsymbol{s}_{\ell,n,m}$. Since the weights $\boldsymbol{\theta}_\ell$ are shared across the weight-sharing dimension of size $R$, we can write the $r$-th row of $\boldsymbol{S}_{\ell,n}$ as $\boldsymbol{s}_{\ell,n,r} = \boldsymbol{W}_\ell \boldsymbol{a}_{\ell,n,r}$ and we have $\boldsymbol{J}_{\boldsymbol{\theta}_\ell} \boldsymbol{s}_{\ell,n,r} = \boldsymbol{a}_{\ell,n,r}^{\mathsf{T}} \otimes \mathbf{I}_{P_{\ell,\text{out}}}$, as for a regular linear layer. We denote $\boldsymbol{b}_{\ell,n,r,m} := \boldsymbol{J}_{\boldsymbol{s}_{\ell,n,m}} f_{\boldsymbol{\theta}}(\boldsymbol{x}_n)_r^{\mathsf{T}}$.

Hence, we have $\boldsymbol{J}_{\boldsymbol{\theta}_\ell}(\boldsymbol{x}_n)_r^\mathsf{T} = \sum_{m=1}^{R} \boldsymbol{a}_{\ell,n,m} \otimes \boldsymbol{b}_{\ell,n,r,m}$. Deriving the approximation the same way as we would with $N$ examples is not possible, since we cannot directly write each of the $N \cdot R$ loss terms as a Kronecker product without any approximation. The Jacobians $\boldsymbol{J}_{\boldsymbol{\theta}_\ell}(\boldsymbol{x}_n)_r$ could be approximated with a Kronecker product of sums, but this requires access to $\sum_{m=1}^{R} \boldsymbol{b}_{\ell,n,r,m}$ and would not be exact in the simple settings we consider later. In contrast, what can be implemented in practice without additional backward passes and what has been used for convolutional neural networks (Grosse & Martens, 2016) and language transformers (Grosse et al., 2023; Osawa et al., 2022; Pauloski et al., 2021; Zhang et al., 2019a) is

$$
\begin{aligned}
\mathbf{GGN}(\boldsymbol{\theta}_\ell) &= \sum_{n=1}^{N} \sum_{r=1}^{R} \left( \sum_{m=1}^{R} \boldsymbol{a}_{\ell,n,m} \otimes \boldsymbol{b}_{\ell,n,r,m} \right) \boldsymbol{\Lambda}(f_{\boldsymbol{\theta}}(\boldsymbol{x}_n)_r) \left( \sum_{m'=1}^{R} \boldsymbol{a}_{\ell,n,m'}^\mathsf{T} \otimes \boldsymbol{b}_{\ell,n,r,m'}^\mathsf{T} \right) \\
&\approx \sum_{n=1}^{N} \sum_{r=1}^{R} \sum_{m=1}^{R} \left( \boldsymbol{a}_{\ell,n,m} \otimes \boldsymbol{b}_{\ell,n,r,m} \right) \boldsymbol{\Lambda}(f_{\boldsymbol{\theta}}(\boldsymbol{x}_n)_r) \left( \boldsymbol{a}_{\ell,n,m}^\mathsf{T} \otimes \boldsymbol{b}_{\ell,n,r,m}^\mathsf{T} \right),
\end{aligned}
\tag{6}
$$

where all terms with $m \neq m'$ are ignored.[4] Consequently, we can apply the regular K-FAC approximation over $N \cdot R$ terms instead of the usual $N$ terms, resulting in what we call the *K-FAC-expand* approximation:

---

**K-FAC-expand**

$$
\mathbf{G\hat{G}N}_{\boldsymbol{\theta}_\ell}^{\mathrm{expand}} := \underbrace{\left[ \frac{1}{NR} \sum_{n=1}^{N} \sum_{m=1}^{R} \boldsymbol{a}_{\ell,n,m} \boldsymbol{a}_{\ell,n,m}^\mathsf{T} \right]}_{=\boldsymbol{A}_\ell} \otimes \underbrace{\left[ \sum_{n=1}^{N} \sum_{m=1}^{R} \sum_{r=1}^{R} \boldsymbol{b}_{\ell,n,r,m} \boldsymbol{\Lambda}(f_{\boldsymbol{\theta}}(\boldsymbol{x}_n)_r) \boldsymbol{b}_{\ell,n,r,m}^\mathsf{T} \right]}_{=\boldsymbol{B}_\ell}
$$

$$\tag{7}$$

---

In the context of convolutions, this approximation has been derived by explicitly stating the assumptions on the activations and pre-activation derivatives (Grosse & Martens, 2016).

For deep linear networks without any weight-sharing layers, K-FAC is known to be exact assuming a Gaussian likelihood (Bernacchia et al., 2018). While this holds for the full GGN/Fisher, we only focus on the block-diagonal case here. To motivate K-FAC-expand, we want to show that the same also holds for deep linear networks with weight-sharing in the expand setting. However, it does not even hold for simplistic transformer models since the dot-product attention mechanism in transformers directly correlates elements across the weight-sharing dimension; we discuss this in more detail in Appendix B. A deep linear network is defined as a model of the form

$$
f_{\boldsymbol{\theta}}(\boldsymbol{x}) = \boldsymbol{W}_L \ldots \boldsymbol{W}_\ell \ldots \boldsymbol{W}_1 \boldsymbol{x} = \boldsymbol{W} \boldsymbol{x},
\tag{8}
$$

where $\boldsymbol{x} \in \mathbb{R}^D$ and $\boldsymbol{W}_L \in \mathbb{R}^{C \times P_{L,\mathrm{in}}}$, $\boldsymbol{W}_\ell \in \mathbb{R}^{P_{\ell,\mathrm{out}} \times P_{\ell,\mathrm{in}}}$ (with $P_{\ell,\mathrm{in}} = P_{\ell-1,\mathrm{out}}$), and $\boldsymbol{W}_1 \in \mathbb{R}^{P_{1,\mathrm{out}} \times D}$. Decomposing a single weight matrix $\boldsymbol{W}$ into $L$ separate ones is a common framework for theoretical analysis since it creates nonlinear training dynamics for gradient-based training algorithms, while still having analytical solutions (Bernacchia et al., 2018; Saxe et al., 2014). For an input $\boldsymbol{X} \in \mathbb{R}^{R \times D}$ to this type of network, we can show:

**Proposition 1** (**Exactness of K-FAC-expand for deep linear network in the expand setting**). *For layer $\ell$ of a deep linear network defined as in Equation (8) and a Gaussian likelihood with p.d. covariance matrix $\boldsymbol{\Sigma} \in \mathbb{R}^{C \times C}$, K-FAC-expand is exact in the expand setting.*

## 3.3 K-FAC-reduce

The reduce setting is characterised by a loss with just $N$ loss terms, i.e. $\mathcal{L}_{\mathrm{reduce}}(f_{\boldsymbol{\theta}}, \mathcal{D}) := -\sum_{n=1}^{N} \log p(\boldsymbol{y}_n | f_{\boldsymbol{\theta}}(\boldsymbol{x}_n))$, and thus the weight-sharing dimension must have been reduced somewhere in the forward pass of the neural network $f_{\boldsymbol{\theta}}$. A typical instance where this type of loss is used in a model with linear weight-sharing layers is image classification with a convolutional neural network or vision transformer.

---

[4]Although the work on transformers does not scale by $1/R$ in Equation (7), see Appendix B.2.1 for details.

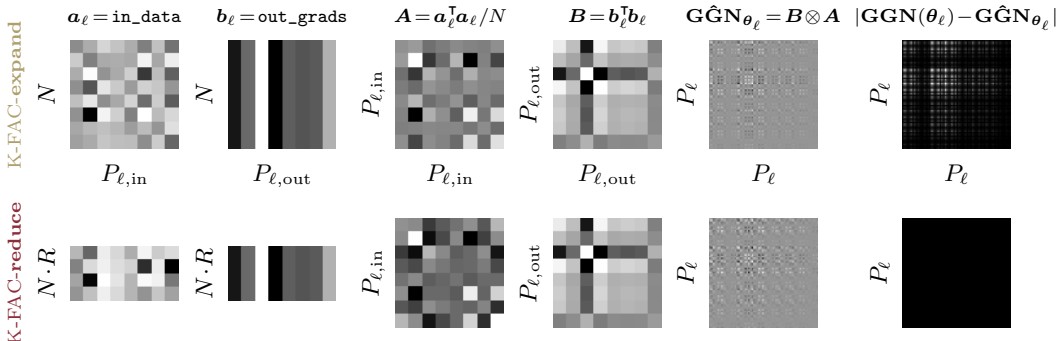

Figure 2: **Visualisation of K-FAC-expand and K-FAC-reduce in the reduce setting.** This is similar to Figure 1, but for the reduce setting, where K-FAC-reduce is exact and K-FAC-expand is not (Section 3.3).

Since $\boldsymbol{A}_{\ell,n} \in \mathbb{R}^{R \times P_{\ell,\text{in}}}$ is now a matrix, we have $\boldsymbol{S}_{\ell,n} = \boldsymbol{A}_{\ell,n} \boldsymbol{W}_\ell^\intercal \in \mathbb{R}^{R \times P_{\ell,\text{out}}}$. Hence, $\boldsymbol{J}_{\boldsymbol{\theta}_\ell} \boldsymbol{S}_{\ell,n}$ and $\boldsymbol{J}_{\boldsymbol{S}_{\ell,n}} f_{\boldsymbol{\theta}}(\boldsymbol{x}_n)$ are now both multi-dimensional arrays. Luckily, we can avoid dealing with this directly by writing $\boldsymbol{J}_{\boldsymbol{\theta}_\ell} f_{\boldsymbol{\theta}}(\boldsymbol{x}_n) = \sum_{r=1}^R \boldsymbol{J}_{\boldsymbol{s}_{\ell,n,r}} f_{\boldsymbol{\theta}}(\boldsymbol{x}_n) \boldsymbol{J}_{\boldsymbol{\theta}_\ell} \boldsymbol{s}_{\ell,n,r}$, where $\boldsymbol{s}_{\ell,n,r} \in \mathbb{R}^{P_{\ell,\text{out}}}$ is the $r$-th row of $\boldsymbol{S}_{\ell,n}$ and $\boldsymbol{s}_{\ell,n,r} = \boldsymbol{W}_\ell \boldsymbol{a}_{\ell,n,r}$. Using this equivalence, we can approximate the (transposed) Jacobians in the GGN for layer $\ell$ as

$$\boldsymbol{J}_{\boldsymbol{\theta}_\ell} f_{\boldsymbol{\theta}}(\boldsymbol{x}_n)^\intercal = \sum_{r=1}^R \boldsymbol{a}_{\ell,n,r} \otimes \boldsymbol{b}_{\ell,n,r} \approx \frac{1}{R} \sum_{r=1}^R \boldsymbol{a}_{\ell,n,r} \otimes \sum_{r=1}^R \boldsymbol{b}_{\ell,n,r}. \tag{9}$$

Here, we have approximated the sum of Kronecker products with a Kronecker product of sums over $R$ terms of each of the $N$ per-input Jacobians. This approximation has been proposed in Tang et al. (2021) to improve the efficiency of their proposed K-FAC variation for convolutions and in a different context for invariance learning with deep neural networks via differentiable Laplace approximations (Immer et al., 2022). We can apply the same approximation as usual to the sum over the $N$ data points and call the resulting final approximation *K-FAC-reduce* and can write it explicitly as

---

**K-FAC-reduce**

$$\mathbf{G}\hat{\mathbf{G}}\mathbf{N}_{\boldsymbol{\theta}_\ell}^{\text{reduce}} :=$$
$$\underbrace{\left[ \frac{1}{NR^2} \sum_{n=1}^N \left( \sum_{r=1}^R \boldsymbol{a}_{\ell,n,r} \right) \left( \sum_{r=1}^R \boldsymbol{a}_{\ell,n,r}^\intercal \right) \right]}_{=\hat{\boldsymbol{A}}_\ell} \otimes \underbrace{\left[ \sum_{n=1}^N \left( \sum_{r=1}^R \boldsymbol{b}_{\ell,n,r} \right) \boldsymbol{\Lambda}(f_{\boldsymbol{\theta}}(\boldsymbol{X}_n)) \left( \sum_{r=1}^R \boldsymbol{b}_{\ell,n,r}^\intercal \right) \right]}_{=\hat{\boldsymbol{B}}_\ell}. \tag{10}$$

---

To ensure that K-FAC-reduce is exact in the simple setting of Proposition 1, we have to choose an appropriate aggregation function $z : \mathbb{R}^{R \times P_{\ell,\text{out}}} \to \mathbb{R}^{P_{\ell,\text{out}}}$. A simple and relevant case for which this holds is a *scaled sum*, i.e. $z(\boldsymbol{S}_{\ell,n}) = c \sum_{r=1}^R \boldsymbol{s}_{\ell,n,r}$ with $c \in \mathbb{R}$. Both vision transformers and convolutional neural networks with average pooling use scaled sums as the aggregation function (with $c = 1/R$). With this, we can state the corresponding statement to Proposition 1.

**Proposition 2 (Exactness of K-FAC-reduce for deep linear network in the reduce setting).** *For layer $\ell$ of a deep linear network (Equation (8)), a Gaussian likelihood with p.d. covariance matrix $\boldsymbol{\Sigma} \in \mathbb{R}^{C \times C}$, and a scaled sum aggregation function, K-FAC-reduce is exact in the reduce setting.*

Notably, for a deep linear model with convolution layers and average pooling, K-FAC reduce will be exact under the assumptions of Proposition 2, whereas the approximation proposed in the literature (Grosse & Martens, 2016) (which corresponds to K-FAC-expand) is not. Moreover, the calculation of $\boldsymbol{A}_\ell$ costs $\mathcal{O}(NRP_{\ell,\text{in}}^2)$ for K-FAC-expand and only $\mathcal{O}(NP_{\ell,\text{in}}(P_{\ell,\text{in}} + R))$ for $\hat{\boldsymbol{A}}_\ell$ and K-FAC-reduce. The same holds for $\boldsymbol{B}_\ell$ and $\hat{\boldsymbol{B}}_\ell$, with complexities of $\mathcal{O}(NRCP_{\ell,\text{out}}(C + P_{\ell,\text{out}}))$ and $\mathcal{O}(NCP_{\ell,\text{out}}(C + P_{\ell,\text{out}} + R))$. The space complexity of K-FAC's overhead also differs by a factor of $R$, with $\mathcal{O}(NR(P_{\ell,\text{in}} + P_{\ell,\text{out}}))$ for K-FAC-expand and $\mathcal{O}(N(P_{\ell,\text{in}} + P_{\ell,\text{out}}))$ for K-FAC-reduce.

# 4 Experiments

In Section 4.1, we empirically validate the speed up of K-FAC-reduce compared to K-FAC-expand in line with their computational complexities with a Wide ResNet on CIFAR-10 and show how this scales with the batch size. Additionally, in Sections 4.2 and 4.3, we compare the two approximations based on their downstream optimisation performance. We validate that both K-FAC variations can reduce the number of required steps to achieve a fixed validation target compared to a well-tuned first-order reference optimiser. For this, we use a GNN and a vision transformer (ViT), two of the examples in Section 2.2. We leverage the codebase and validation metric targets of the MLCommons `algorithmic-efficiency` (AlgoPerf) repository[5] (Dahl et al., 2023), and use their target-setting runs as reference runs. Both K-FAC variations use the same learning rate schedule as those target-setting runs but the warmup and expected number of steps are multiplied by 0.75 to account for the fact that second-order methods tend to require fewer steps. Additionally, we tune the learning rate and damping for K-FAC via random search and set all other hyperparameters to the same values as the reference target-setting run. To exclude the adjusted learning rate schedule and the additional tuning of the learning rate as confounding factors, we additionally applied the same tuning to the reference runs; however, none of these runs hit the target validation performance. Despite these considerations, the experiments in Sections 4.2 and 4.3 are *not* meant to demonstrate that K-FAC is a superior training algorithm (see Appendix C.1). However, the reference runs allow us to put the training performances of both K-FAC flavours into perspective and also indicate the potential of extending second-order methods like K-FAC to modern neural network architectures Lastly, to demonstrate a use case of K-FAC other than as a preconditioner, we automatically learn the weight decay hyperparameter by optimising a Laplace approximation of the marginal likelihood in Section 4.4. Our K-FAC implementation leverages the `ASDL` package (Osawa et al., 2023). See Appendix C for more details on the experiments and additional results.

## 4.1 Update step speed with K-FAC-expand and K-FAC-reduce

To empirically validate the smaller computational complexity of K-FAC-reduce compared to K-FAC-expand, we time a single preconditioned gradient update step for a Wide ResNet on CIFAR-10 with both approximations and five different batch sizes on an NVIDIA V100 GPU (Table 1). As expected based on the complexities of the two approximations, we see increasing gains in speed with increasing batch size. For the largest tested batch size, 2048, K-FAC-reduce is faster than K-FAC-expand is for

Table 1: Timing [s] of an update step with the two K-FAC variants for a Wide ResNet on CIFAR-10.

| K-FAC | Batch size | | | | |
|---|---|---|---|---|---|
| | 128 | 256 | 512 | 1024 | 2048 |
| expand | 0.24 | 0.38 | 0.75 | 1.36 | OOM |
| reduce | 0.17 | 0.24 | 0.43 | 0.63 | 1.17 |

half the batch size. Moreover, K-FAC-expand runs out-of-memory (OOM) at a batch size of 2048. Notably, we implement both approximations with the naive unfold operation (c.f. Section 2.2), but K-FAC-reduce could be implemented even more efficiently by never explicitly allocating memory for the full unfolded layer input, which has been shown to lead to a further speed-up of about $4.5\times$ (Dangel, 2023). To highlight the speed difference of the two K-FAC variations for language models, we compute a K-FAC GGN approximation for `nanoGPT` (Karpathy, 2023), a popular implementation of GPT-2 (Radford et al., 2019), on the full DART dataset (Nan et al., 2021). K-FAC-reduce is significantly faster than K-FAC-expand, which is currently used for transformers, only taking 70% of the time of K-FAC-expand. We expect this discrepancy to become even more pronounced for larger batch sizes and sequence lengths.

## 4.2 Graph neural network on ogbg-molpcba

We use a basic GraphNetwork (Battaglia et al., 2018) with multi-layer perceptrons as the update functions (c.f. Appendix A) and train it on the `ogbg-molpcba` dataset (Hu et al., 2020) (Figure 3). The target validation metric is a mean average precision (mAP) of 0.28098. The reference algorithm is SGD with Nesterov momentum, with the tuned hyperparameters from the `AlgoPerf` target-setting runs. For both K-FAC variations, the statistics and preconditioner are updated every 10 iterations. We report the mean and standard error for five runs with different random seeds. The reference run reaches the target in $31{,}887 \pm 2{,}070$ steps, whereas K-FAC-expand takes $16{,}106 \pm 863$ and K-FAC-

---

[5] https://bit.ly/algoperf

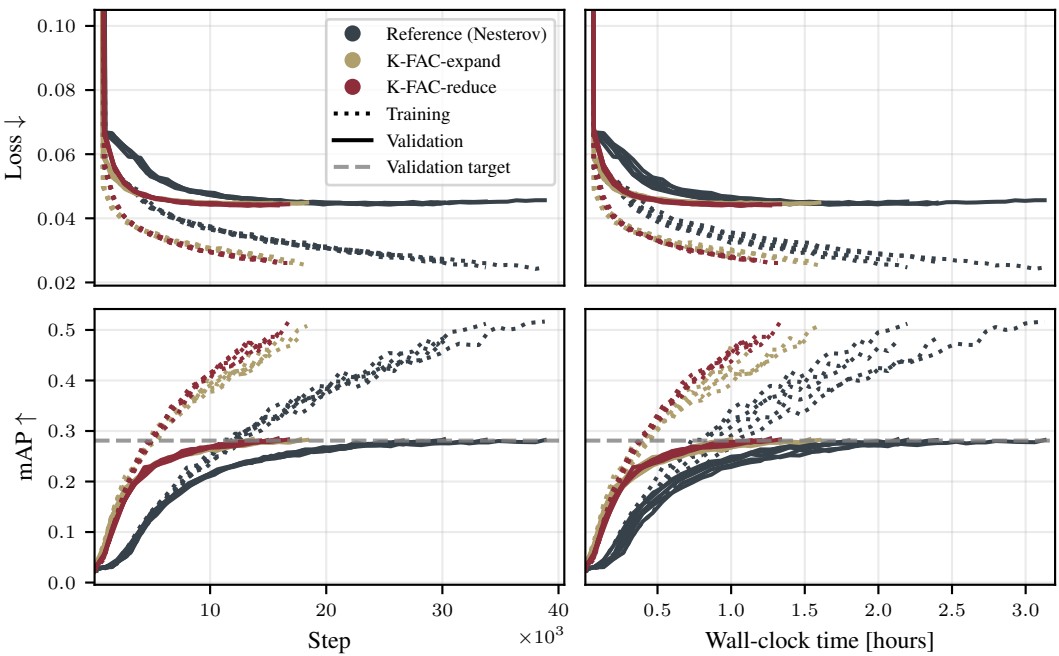

Figure 3: **Training results for a graph neural network on ogbg-molpcba.** Both K-FAC variations require $\approx 50\,\%$ of the steps which almost directly translates into reduced wall-clock time. The K-FAC statistics are updated every 10 steps. We show runs with five different random seeds for each method.

reduce $15{,}566 \pm 429$ steps, about $51\,\%$ and $49\,\%$ of the reference run, respectively. In wall-clock time, the reference run takes about $2.33 \pm 0.22$ hours, whereas K-FAC-expand takes $1.32 \pm 0.11$ and K-FAC-reduce $1.23 \pm 0.03$ hours, about $57\,\%$ and $53\,\%$ of the reference run, respectively. The reduced number of steps almost directly translates into reduced wall-clock time since the update step time is dominated by the data pipeline. K-FAC-reduce appears to be slightly faster in steps and wall-clock time than K-FAC-expand, but not significantly. Moreover, the runs with K-FAC-reduce have lower variance.

### 4.3 Vision transformer on ImageNet

We train a ViT (Dosovitskiy et al., 2021) on LSVRC-2012 ImageNet (Russakovsky et al., 2015) and use NAdamW (Loshchilov & Hutter, 2019) with the hyperparameters from the `AlgoPerf` target-setting runs as the reference run (Figure 4); each run is repeated with three different random seeds. For K-FAC-expand and K-FAC-reduce, we update the K-FAC statistics and preconditioner every 50 iterations. The reference run reaches the target validation accuracy of $0.77309$ in $122{,}139 \pm 1119$ steps or about $26.10 \pm 0.24$ hours. K-FAC-expand only takes $87{,}464 \pm 245$ steps or $19.57 \pm 0.13$ hours, and K-FAC-reduce $92{,}550 \pm 731$ steps or $20.28 \pm 0.00$ hours. This corresponds to about $72\,\%$ of the steps and $75\,\%$ of the time of the reference run for K-FAC-expand and $76\,\%$ of the steps and $78\,\%$ of the reference run's time for K-FAC-reduce. While the difference in speed between both K-FAC variations does become less relevant due to the infrequent K-FAC updates, one update step with K-FAC-expand takes about $1.36\times$ as much as with K-FAC-reduce, which has a significant impact on the runtime when the K-FAC statistics are computed more frequently; see Figure 5 for a run where the update is performed every step. While K-FAC-expand requires slightly fewer steps and wall-clock time than K-FAC-reduce here, the difference does not appear significant.

### 4.4 K-FAC for automatic hyperparameter selection via marginal likelihood optimisation

K-FAC has also been used for marginal likelihood maximisation in Bayesian neural networks using Laplace approximations (Daxberger et al., 2021; Immer et al., 2021). In particular, K-FAC can be used to obtain a scalable Laplace approximation and optimise its marginal likelihood with respect to regularisation parameters, e.g., weight decay, during training.

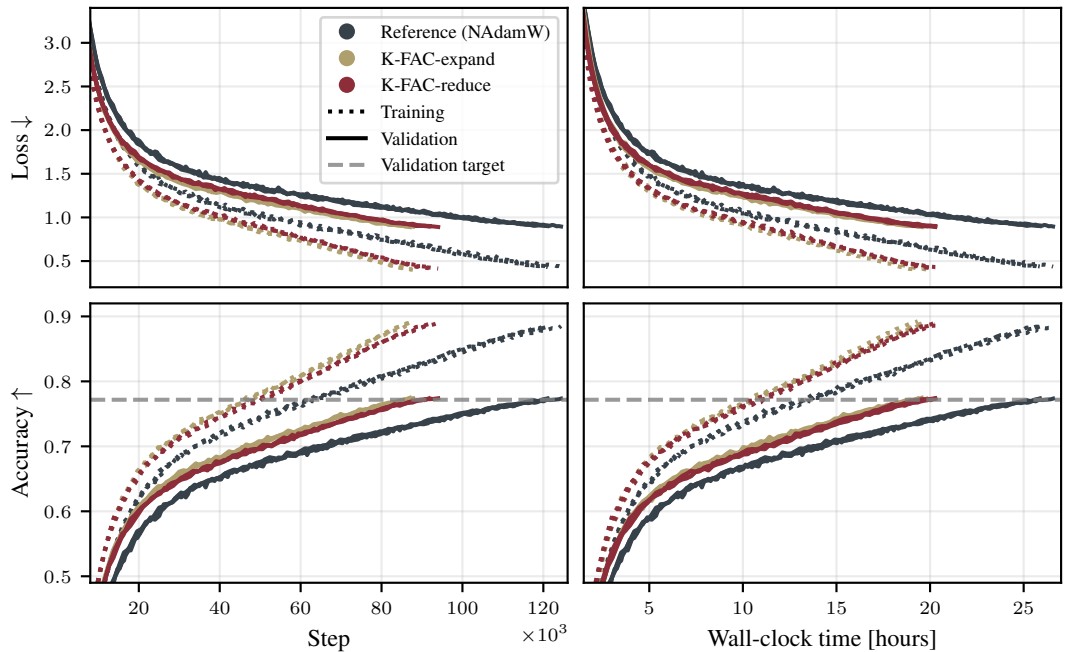

Figure 4: **Training results for a vision transformer on ImageNet.** The K-FAC statistics are updated every 50 steps. Due to amortising the costs, the reduced number of steps to the target translates into reduced wall-clock time. We show runs with three different random seeds for each method.

Therefore, we compare K-FAC-reduce and K-FAC-expand in this setting following the setup of Daxberger et al. (2021) on CIFAR-10 with a Wide ResNet (Zagoruyko & Komodakis, 2016) and optimise a weight decay parameter per layer every five epochs during 100 epochs of training. Table 2 shows that K-FAC-reduce is significantly faster than K-FAC-expand in this setting, but K-FAC-expand provides better test negative log likelihoods (NLL) in both settings.

Table 2: Performance of K-FAC variants for marginal likelihood maximisation on CIFAR-10 with a Wide ResNet with and without data augmentation (DA).

| K-FAC | DA | NLL ↓ | Acc [%] ↑ | Time [%] ↓ |
|---|---|---|---|---|
| expand | ✗ | 0.42 | 88.9 | 100 |
| | ✓ | 0.24 | 92.5 | |
| reduce | ✗ | 0.70 | 86.7 | 50.5 |
| | ✓ | 0.35 | 93.5 | |

## 5 Discussion and conclusion

We have leveraged the idea of linear weight-sharing layers to generalise K-FAC to a broader range of architectures. This resulted in two distinct flavours of the approximation, which are exact for simple cases with deep linear networks in their respective setting. Maybe surprisingly, we see little difference between the two K-FAC variations in terms of their downstream optimisation performance. Moreover, in some settings, the overhead of K-FAC can be amortised or is negligible. This is, for example, the case when 1) the data loading dominates the algorithm's update step time (c.f. the GNN experiment), 2) when the K-FAC update frequency can be reduced sufficiently (c.f. the ViT experiment), or 3) when underutilised resources can be leveraged to compute K-FAC updates, for example when using pipeline parallelism for language models (Osawa et al., 2022). In these settings, the improvement in wall-clock time of K-FAC-reduce compared to K-FAC-expand is rendered insignificant. However, in applications where the K-FAC overhead still significantly impacts the runtime, e.g. in the online weight decay selection experiment in Section 4.4, or when there are memory constraints (c.f. Section 4.1), K-FAC-reduce can greatly reduce the runtime compared to the previously used K-FAC-expand. Some challenges remain when applying K-FAC in modern deep learning settings. For example, the memory overhead can be prohibitive, even for K-FAC-reduce, which could be addressed by sparse structured Kronecker factors (Grosse et al., 2023; Zhang et al., 2018). Also, numerical instabilities might arise in low-precision settings due to the need for matrix inversions or decompositions; this could be addressed by inverse-free methods relying on the K-FAC approximation (Lin et al., 2023).

## Acknowledgements

The authors thank Kazuki Osawa for fruitful discussions and support with using and extending the `ASDL` library. The authors also thank Wu Lin for pointing out a mistake in the original equation of the K-FAC-expand approximation. Finally, the authors thank Felix Dangel for helpful comments on this manuscript. Runa Eschenhagen is supported by ARM and the Cambridge Trust. Alexander Immer is funded by the Max Planck ETH Center for Learning Systems (CLS). Richard E. Turner is supported by Google, Amazon, ARM, Improbable and EPSRC grant EP/T005386/1. Frank Schneider and Philipp Hennig are funded by the Deutsche Forschungsgemeinschaft (DFG, German Research Foundation) in the frame of the priority programme SPP 2298 "Theoretical Foundations of Deep Learning" - HE 7114/5-1, as well as the DFG Cluster of Excellence "Machine Learning - New Perspectives for Science", EXC 2064/1, project number 390727645; the German Federal Ministry of Education and Research (BMBF) through the Tübingen AI Center (FKZ: 01IS18039A); and funds from the Ministry of Science, Research and Arts of the State of Baden-Württemberg.

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

# A Weight-sharing in graph neural networks

In this section, we expand on the third example in Section 2.2 on graph neural networks (GNNs) and show how they use linear layers with weight-sharing.

**Graph convolutional network for node classification.** A popular type of GNN is called graph convolutional network (Kipf & Welling, 2016, GCN). A GCN defines a convolution operation on graph structures, by repeatedly aggregating feature information over the neighbourhood of a node. As a regular convolutional neural network, it also uses weight-sharing; see Liu et al. (2020) for a comprehensive discussion on weight-sharing in GCNs. However, in contrast to the other models presented here, they utilise a slightly different type of weight-sharing, which will become apparent in Equation (12). Nevertheless, we briefly mention this case here, since the only work on K-FAC for GNNs has been on this model architecture and we will explicitly show how K-FAC was applied in this case in Appendix B.3.2; this relies on the notation introduced here.

A graph is defined as $\mathcal{G} := (\mathcal{V}, \mathcal{E})$, where $\mathcal{V}$ is the set of $N$ nodes and $\mathcal{E}$ the set of edges. The edges can be encoded relative to the nodes in an adjacency matrix $\boldsymbol{C} \in \mathbb{R}^{N \times N}$ with $C_{ij} = 0$ if there is no edge and $C_{ij} = 1$ if there is one. Typically, they are used for node and graph classification tasks. Here, we focus on node classification, e.g. classifying scientific publications which are represented as nodes in a citation network into topics (Sen et al., 2008).

The $\ell$-th GCN layer is defined as

$$f_{\boldsymbol{\theta}_\ell}(\boldsymbol{X}) = \phi(\hat{\boldsymbol{C}} \boldsymbol{X} \boldsymbol{W}_\ell^\mathsf{T}) \tag{11}$$

which is identical to a regular dense linear layer from Section 2.2, but the input matrix $\boldsymbol{X} \in \mathbb{R}^{N \times P_{\ell,\text{in}}}$, which has the $N$ node features $\boldsymbol{x}_n$ of size $P_{\ell,\text{in}}$ stacked in the rows, is first transformed by the normalised adjacency matrix $\hat{\boldsymbol{C}} := (\boldsymbol{D} + \mathbf{I}_N)^{-\frac{1}{2}}(\boldsymbol{C} + \mathbf{I}_N)(\boldsymbol{D} + \mathbf{I}_N)^{-\frac{1}{2}}$, where $\boldsymbol{D}$ is the diagonal node degree matrix of the graph and $\mathbf{I}_N$ is the $N \times N$ identity matrix.

Defining

$$\tilde{\boldsymbol{x}}_n := \sum_{j=1}^{N} \hat{\boldsymbol{C}}_{nj} \boldsymbol{x}_j = \sum_{j \in \mathcal{N}(n)} \hat{\boldsymbol{C}}_{nj} \boldsymbol{x}_j, \tag{12}$$

we can express the forward pass for a single node and layer as

$$f_{\boldsymbol{\theta}_\ell}(\tilde{\boldsymbol{x}}_n) = \phi(\boldsymbol{W}_\ell \tilde{\boldsymbol{x}}_n), \tag{13}$$

where $\mathcal{N}(n) := \{j \in \{1, \ldots, N\} | \hat{\boldsymbol{C}}_{nj} \neq 0\}$ is the neighbourhood of the node with index $n$. Notably, the forward pass for a single node $\boldsymbol{x}_n$ depends on its neighbourhood, i.e. we cannot express the forward pass for the node without access to the feature information of the nodes in its neighbourhood $\mathcal{N}(n)$. Moreover, we can now see that the forward pass through the linear layer, i.e. the matrix multiplication of the weight matrix $\boldsymbol{W}_\ell$ with the transformed input $\tilde{\boldsymbol{x}}_n$, does not need the notion of weight-sharing anymore, in the sense, that we do not need a batched matrix-vector product over a weight-sharing dimension. This is because we aggregate over each node's neighbourhood, over which the weights are shared, *before* the matrix-vector product. Hence, in contrast to the GraphNetwork introduced in the next paragraph, this model does not require special consideration when applying K-FAC (c.f. setting (i) in Section 3.1).

**GraphNetwork for graph classification.** One more general formulation of a GNN is an instance of the GraphNetwork introduced in Battaglia et al. (2018). The GraphNetwork in its general form takes a graph $\mathcal{G} = (\boldsymbol{u}, \mathcal{V}, \mathcal{E})$ where $\boldsymbol{u} \in \mathbb{R}^{D_u}$ are the global features of the graph, and $\mathcal{V}$ and $\mathcal{E}$ are the sets of nodes and edges, respectively, just as before. We can also write the $i$-th graph of a dataset of $N$ graphs as a 5-tuple $\mathbb{X}_n^G := (\boldsymbol{x}_n^u, \boldsymbol{X}_n^V, \boldsymbol{X}_n^E, \boldsymbol{r}_n, \boldsymbol{s}_n)$, with global features $\boldsymbol{x}_n^u \in \mathbb{R}^{D_u}$, node features $\boldsymbol{X}_n^V \in \mathbb{R}^{N_n^V \times D_V}$, and edge features $\boldsymbol{X}_n^E \in \mathbb{R}^{N_n^E \times D_E}$ for all $n = 1, \ldots, N$. The two vectors $\boldsymbol{r}_n \in \mathbb{R}^{N_n^E}$ and $\boldsymbol{s}_n \in \mathbb{R}^{N_n^E}$ contain the indices of the receiving and sending nodes of each edge, respectively. Using these indices, we define $\boldsymbol{X}_{n,\boldsymbol{r}_n}^V \in \mathbb{R}^{N_n^E \times D_V}$ and $\boldsymbol{X}_{n,\boldsymbol{s}_n}^V \in \mathbb{R}^{N_n^E \times D_V}$ which contain the node features $\boldsymbol{X}_n^V$ at indices $\boldsymbol{s}_n$ and $\boldsymbol{r}_n$, respectively. Note, that these graph inputs unfortunately cannot trivially be batched by stacking them, since the number of nodes $N_n^V$ or edges $N_n^E$ are not necessarily the same for all $n \in \{1, \ldots, N\}$.

A GraphNetwork block updates the 3-tuple $(\boldsymbol{x}_n^u, \boldsymbol{X}_n^V, \boldsymbol{X}_n^E)$ by using three update functions $\phi$,

$$
\begin{aligned}
\boldsymbol{X}_n^E &\leftarrow \phi^E(\boldsymbol{X}_n^E, \boldsymbol{X}_{n,\boldsymbol{r}_n}^V, \boldsymbol{X}_{n,\boldsymbol{s}_n}^V, \boldsymbol{x}_n^u) \\
\boldsymbol{X}_n^V &\leftarrow \phi^V(\boldsymbol{X}_n^V, \tilde{\boldsymbol{X}}_n^E, \boldsymbol{x}_n^u) \\
\boldsymbol{x}_n^u &\leftarrow \phi^u(\boldsymbol{x}_n^u, \bar{\boldsymbol{X}}_n^V, \bar{\boldsymbol{X}}_n^E),
\end{aligned}
\tag{14}
$$

and three permutation-invariant aggregation functions $\rho$

$$
\begin{aligned}
\tilde{\boldsymbol{X}}_n^E &\leftarrow \rho^{E \to V}(\boldsymbol{X}_n^E) \\
\bar{\boldsymbol{X}}_n^E &\leftarrow \rho^{E \to u}(\boldsymbol{X}_n^E) \\
\bar{\boldsymbol{X}}_n^V &\leftarrow \rho^{V \to u}(\boldsymbol{X}_n^V).
\end{aligned}
\tag{15}
$$

Examples of these aggregation functions include element-wise summation, mean, or maximum.

One forward pass through a GraphNetwork block corresponds to the following steps, where each step is executed for all $n \in \{1, \dots, N\}$:

1. Update edges $\boldsymbol{X}_n^E$ with $\phi^E(\boldsymbol{X}_n^E, \boldsymbol{X}_{n,\boldsymbol{r}_n}^V, \boldsymbol{X}_{n,\boldsymbol{s}_n}^V, \boldsymbol{x}_n^u)$.

2. Aggregate updated edges over all nodes in $\tilde{\boldsymbol{X}}_n^E \in \mathbb{R}^{N_n^V \times D_E}$ using $\rho^{E \to V}(\boldsymbol{X}_n^E)$.

3. Update nodes $\boldsymbol{X}_n^V$ using $\phi^V(\boldsymbol{X}_n^V, \tilde{\boldsymbol{X}}_n^E, \boldsymbol{x}_n^u)$.

4. Aggregate updated edges over all graphs in $\bar{\boldsymbol{X}}_n^E \in \mathbb{R}^{D_E}$ using $\rho^{E \to u}(\boldsymbol{X}_n^E)$.

5. Aggregate updated nodes over all graphs in $\bar{\boldsymbol{X}}_n^V \in \mathbb{R}^{D_V}$ using $\rho^{V \to u}(\boldsymbol{X}_n^V)$.

6. Update global features $\boldsymbol{x}_n^u$ with $\phi^u(\boldsymbol{x}_n^u, \bar{\boldsymbol{X}}_n^V, \bar{\boldsymbol{X}}_n^E)$.

In this work, we consider graph classification; for example, molecules can be represented as graphs and we could classify them according to some chemical property (e.g. the ogbg-molpcba dataset used in Section 4.2). We specifically consider a GraphNetwork instance with simple MLPs for all update functions $\phi$ and an element-wise sum for the aggregation functions $\rho$. Moreover, multiple of these GraphNetwork blocks can be stacked on top of each other. To classify the input graphs, an MLP is applied to the global features $\boldsymbol{x}_n^u$ after they are updated by the last GraphNetwork block.

To be more precise, the update functions are in this case specified as

$$
\begin{aligned}
\phi^E(\boldsymbol{X}_n^E, \boldsymbol{X}_{n,\boldsymbol{r}_n}^V, \boldsymbol{X}_{n,\boldsymbol{s}_n}^V, \boldsymbol{x}_n^u) &:= \mathrm{concat}(\boldsymbol{X}_n^E, \boldsymbol{X}_{n,\boldsymbol{r}_n}^V, \boldsymbol{X}_{n,\boldsymbol{s}_n}^V, \mathrm{repeat}_{N_n^E}(\boldsymbol{x}_n^u)) \boldsymbol{W}^{E\mathsf{T}} \\
\phi^V(\boldsymbol{X}_n^V, \tilde{\boldsymbol{X}}_n^E, \boldsymbol{x}_n^u) &:= \mathrm{concat}(\boldsymbol{X}_n^V, \tilde{\boldsymbol{X}}_n^E, \mathrm{repeat}_{N_n^V}(\boldsymbol{x}_n^u)) \boldsymbol{W}^{V\mathsf{T}} \\
\phi^u(\boldsymbol{x}_n^u, \bar{\boldsymbol{X}}_n^V, \bar{\boldsymbol{X}}_n^E) &:= \boldsymbol{W}^u \mathrm{concat}(\boldsymbol{x}_n^u, \bar{\boldsymbol{X}}_n^V, \bar{\boldsymbol{X}}_n^E)
\end{aligned}
\tag{16}
$$

with $\boldsymbol{W}^E \in \mathbb{R}^{D_E \times (D_E + 2D_V + D_u)}$, $\boldsymbol{W}^V \in \mathbb{R}^{D_V \times (D_V + D_E + D_u)}$, and $\boldsymbol{W}^u \in \mathbb{R}^{D_u \times 3D_u}$.

Note, that this is a simplification since in reality, the update functions $\phi$ are MLPs with ReLU activations, layer normalisation (Ba et al., 2016), and dropout (Hinton et al., 2012). Also, we omit the potential bias vectors. However, these components are not relevant for deriving K-FAC for the linear layers within these networks, which is why we can omit them here for simplicity.

Most importantly, we can now observe that this type of GNN shares its weights over each graph's edges and nodes: just as for the attention or convolution operations described in Section 2.2, we apply the transposed weight matrices from the right side of the input of the layers of type $\phi^E$ and $\phi^V$, i.e. for updating the edge and node features. However, since the number of edges $N_n^E$ and the number of nodes $N_n^V$ is not necessarily the same for all $N$ graphs, we now have a weight-sharing dimension of size $R_n$, which depends on the $n$-th input. Also, note that the update function $\phi^u$ used to update the global features $\boldsymbol{x}_n^u$ does not use any weight-sharing, as there is just one feature vector per data point. We have specifically introduced this notation of the inputs to show that the edge and node feature update functions are exactly examples of the concept of a linear weight-sharing layer, as introduced in Section 2.2. This might not be immediately obvious when only looking at the original notation used in Battaglia et al. (2018); therefore, this can be seen as an instructive example for expressing a neural network architecture in terms of linear weight-sharing layers. Consequently, our framework for K-FAC can directly be applied to this architecture, as we show in Appendix B.3.2.

# B Extended derivation and discussion of K-FAC-expand and K-FAC-reduce

## B.1 Background: K-FAC for regular linear layers

Kronecker-Factored Approximate Curvature (Heskes, 2000; Martens & Grosse, 2015, K-FAC) was proposed as an efficient approximation to a neural network's Fisher information matrix. The Fisher information matrix is defined as[6]

$$\boldsymbol{F}(\boldsymbol{\theta}) = -\sum_{n=1}^{N} \mathbb{E}_{\boldsymbol{y} \sim p(\boldsymbol{y}|f_{\boldsymbol{\theta}}(\boldsymbol{x}_n))}[\nabla_{\boldsymbol{\theta}}^2 \log p(\boldsymbol{y}|f_{\boldsymbol{\theta}}(\boldsymbol{x}_n))]$$

$$= \sum_{n=1}^{N} \mathbb{E}_{\boldsymbol{y} \sim p(\boldsymbol{y}|f_{\boldsymbol{\theta}}(\boldsymbol{x}_n))}[\nabla_{\boldsymbol{\theta}} \log p(\boldsymbol{y}|f_{\boldsymbol{\theta}}(\boldsymbol{x}_n))(\nabla_{\boldsymbol{\theta}} \log p(\boldsymbol{y}|f_{\boldsymbol{\theta}}(\boldsymbol{x}_n)))^{\mathsf{T}}]. \tag{17}$$

Notably, the labels $\boldsymbol{y}$ are samples from the predictive distribution of the model and are not the empirical labels from the data. Replacing $\boldsymbol{y}$ with $\boldsymbol{y}_n$ leads to the *empirical* Fisher (EF), which is simply the uncentered covariance of the empirical gradient. While it is commonly used as a replacement for the Fisher (Chaudhari et al., 2017; Graves, 2011; Kingma & Ba, 2015), it can give rise to very different downstream behaviour when used for optimisation (Kunstner et al., 2019).

First, in all of this work, we focus on a layer-wise K-FAC approximation of the Fisher, i.e. it is approximated by a block-diagonal matrix

$$\boldsymbol{F}(\boldsymbol{\theta}) \approx \text{diag}(\boldsymbol{F}(\boldsymbol{\theta}_1), \dots, \boldsymbol{F}(\boldsymbol{\theta}_\ell), \dots, \boldsymbol{F}(\boldsymbol{\theta}_L)) \in \mathbb{R}^{P \times P}, \tag{18}$$

where $\boldsymbol{F}(\boldsymbol{\theta}_\ell) \in \mathbb{R}^{P_\ell \times P_\ell}$ and $\text{diag}(\cdot, \dots, \cdot)$ build a block-diagonal matrix with the input matrices as blocks.

To derive K-FAC, we first note that the pre-activation for layer $\ell$ and the $n$-th data point $\boldsymbol{x}_n$ can be expressed as $\boldsymbol{s}_{\ell,n} = \boldsymbol{W}_\ell \boldsymbol{a}_{\ell,n}$, with $\boldsymbol{W}_\ell \in \mathbb{R}^{P_{\ell,\text{out}} \times P_{\ell,\text{in}}}$ and $\boldsymbol{a}_{\ell,n} \in \mathbb{R}^{P_{\ell,\text{in}}}$, the input to the $\ell$-th layer (or equivalently, the activation of the $\ell-1$-th layer). We have omitted an explicit bias parameter $\boldsymbol{b}_\ell$, since it can always be subsumed in $\boldsymbol{W}_\ell$. Hence, by applying the chain rule, the gradient of the loss w.r.t. the weights of the $\ell$-th layer can be written as $\nabla_{\boldsymbol{W}_\ell} \mathcal{L}(\boldsymbol{y}, f_{\boldsymbol{\theta}}(\boldsymbol{x}_n)) = \nabla_{\boldsymbol{s}_{\ell,n}} \mathcal{L}(\boldsymbol{y}, f_{\boldsymbol{\theta}}(\boldsymbol{x}_n)) \boldsymbol{a}_{\ell,n}^{\mathsf{T}} =: \boldsymbol{g}_{\ell,n} \boldsymbol{a}_{\ell,n}^{\mathsf{T}} \in \mathbb{R}^{P_{\ell,\text{out}} \times P_{\ell,\text{in}}}$.

Using these insights, K-FAC then replaces the sum of expectations over Kronecker products with a Kronecker product of two sums of expectations, i.e.

$$\boldsymbol{F}(\boldsymbol{\theta}_\ell) = \sum_{n=1}^{N} \mathbb{E}_{\boldsymbol{y} \sim p(\boldsymbol{y}|f_{\boldsymbol{\theta}}(\boldsymbol{x}_n))}[\text{vec}(\nabla_{\boldsymbol{W}_\ell} \mathcal{L}(\boldsymbol{y}, f_{\boldsymbol{\theta}}(\boldsymbol{x}_n)))\text{vec}(\nabla_{\boldsymbol{W}_\ell} \mathcal{L}(\boldsymbol{y}, f_{\boldsymbol{\theta}}(\boldsymbol{x}_n)))^{\mathsf{T}}] \tag{19a}$$

$$= \sum_{n=1}^{N} \mathbb{E}_{\boldsymbol{y} \sim p(\boldsymbol{y}|f_{\boldsymbol{\theta}}(\boldsymbol{x}_n))}[\text{vec}(\boldsymbol{g}_{\ell,n} \boldsymbol{a}_{\ell,n}^{\mathsf{T}})\text{vec}(\boldsymbol{g}_{\ell,n} \boldsymbol{a}_{\ell,n}^{\mathsf{T}})^{\mathsf{T}}] \tag{19b}$$

$$= \sum_{n=1}^{N} \mathbb{E}_{\boldsymbol{y} \sim p(\boldsymbol{y}|f_{\boldsymbol{\theta}}(\boldsymbol{x}_n))}[(\boldsymbol{a}_{\ell,n} \otimes \boldsymbol{g}_{\ell,n})(\boldsymbol{a}_{\ell,n}^{\mathsf{T}} \otimes \boldsymbol{g}_{\ell,n}^{\mathsf{T}})] \tag{19c}$$

$$= \sum_{n=1}^{N} \mathbb{E}_{\boldsymbol{y} \sim p(\boldsymbol{y}|f_{\boldsymbol{\theta}}(\boldsymbol{x}_n))}[\boldsymbol{a}_{\ell,n} \boldsymbol{a}_{\ell,n}^{\mathsf{T}} \otimes \boldsymbol{g}_{\ell,n} \boldsymbol{g}_{\ell,n}^{\mathsf{T}}] \tag{19d}$$

$$\approx \underbrace{\left[\frac{1}{N}\sum_{n=1}^{N} \boldsymbol{a}_{\ell,n} \boldsymbol{a}_{\ell,n}^{\mathsf{T}}\right]}_{=:\boldsymbol{A}_\ell} \otimes \underbrace{\left[\sum_{n=1}^{N} \mathbb{E}_{\boldsymbol{y} \sim p(\boldsymbol{y}|f_{\boldsymbol{\theta}}(\boldsymbol{x}_n))}[\boldsymbol{g}_{\ell,n} \boldsymbol{g}_{\ell,n}^{\mathsf{T}}]\right]}_{=:\boldsymbol{G}_\ell}, \tag{19e}$$

where $\boldsymbol{A}_\ell \in \mathbb{R}^{P_{\ell,\text{in}} \times P_{\ell,\text{in}}}$ and $\boldsymbol{G}_\ell \in \mathbb{R}^{P_{\ell,\text{out}} \times P_{\ell,\text{out}}}$. For this derivation, we have used three convenient properties of the Kronecker product (using matrices $\boldsymbol{A}, \boldsymbol{B}, \boldsymbol{C}, \boldsymbol{D}$ with appropriate dimensions): $\text{vec}(\boldsymbol{A}\boldsymbol{B}\boldsymbol{C}) = (\boldsymbol{C}^{\mathsf{T}} \otimes \boldsymbol{A})\text{vec}(\boldsymbol{B})$ and $(\boldsymbol{A} \otimes \boldsymbol{B})^{\mathsf{T}} = \boldsymbol{A}^{\mathsf{T}} \otimes \boldsymbol{B}^{\mathsf{T}}$ for Equation (19c), and $(\boldsymbol{A} \otimes \boldsymbol{B})(\boldsymbol{C} \otimes \boldsymbol{D}) = \boldsymbol{A}\boldsymbol{C} \otimes \boldsymbol{B}\boldsymbol{D}$ for Equation (19d).

---

[6]More generally, it is defined with an expectation over $\boldsymbol{x} \sim p(\boldsymbol{x})$ as well.

We can see that the approximation is exact in the trivial case of a single data point, i.e. $N = 1$. Moreover, it is also exact in the case of a single linear layer or a deep linear network and a Gaussian likelihood (Bernacchia et al., 2018).

K-FAC is more efficient than a naive block-wise approximation because we only have to store and invert two Kronecker factors instead of a larger dense matrix for each layer, which reduces the memory complexity from $\mathcal{O}(P_{\ell,\text{in}}^2 P_{\ell,\text{out}}^2)$ to $\mathcal{O}(P_{\ell,\text{in}}^2 + P_{\ell,\text{out}}^2)$ and the computational complexity of the preconditioning of the gradient with the approximate Fisher from $\mathcal{O}(P_{\ell,\text{in}}^3 P_{\ell,\text{out}}^3)$ to $\mathcal{O}(P_{\ell,\text{in}}^3 + P_{\ell,\text{out}}^3)$, since

$$
\begin{aligned}
\boldsymbol{F}(\boldsymbol{\theta}_\ell)^{-1}\boldsymbol{g}(\boldsymbol{\theta}_\ell) &\approx (\boldsymbol{A}_\ell \otimes \boldsymbol{G}_\ell)^{-1}\boldsymbol{g}(\boldsymbol{\theta}_\ell) \\
&= \text{vec}\left(\boldsymbol{G}_\ell^{-1}\nabla_{\boldsymbol{W}_\ell}\mathcal{L}(\boldsymbol{y}, f_{\boldsymbol{\theta}}(\boldsymbol{x}_n))\boldsymbol{A}_\ell^{-1}\right)
\end{aligned}
\tag{20}
$$

with $\boldsymbol{g}(\boldsymbol{\theta}_\ell) = \text{vec}\left(\nabla_{\boldsymbol{W}_\ell}\mathcal{L}(\boldsymbol{y}, f_{\boldsymbol{\theta}}(\boldsymbol{x}_n))\right)$ and the property $(\boldsymbol{A} \otimes \boldsymbol{B})^{-1} = \boldsymbol{A}^{-1} \otimes \boldsymbol{B}^{-1}$.

Alternatively, we can derive K-FAC for the GGN (Botev et al., 2017), which will recover the same result as for the Fisher in Equation (5) for many common loss functions, as we have learned Section 2.3. We define $\boldsymbol{J}_{\boldsymbol{\theta}_\ell}(\boldsymbol{x}_n) := \boldsymbol{J}_{\boldsymbol{\theta}_\ell}f_{\boldsymbol{\theta}}(\boldsymbol{x}_n) = \boldsymbol{J}_{\boldsymbol{s}_{\ell,n}}f_{\boldsymbol{\theta}}(\boldsymbol{x}_n)\boldsymbol{J}_{\boldsymbol{\theta}_\ell}\boldsymbol{s}_{\ell,n} \in \mathbb{R}^{C \times P_\ell}$ as the Jacobian of the model outputs w.r.t. the parameters of the $\ell$-th layer and $\boldsymbol{\Lambda}(f_{\boldsymbol{\theta}}(\boldsymbol{x}_n)) := \boldsymbol{H}_{f_{\boldsymbol{\theta}}}\mathcal{L}(\boldsymbol{y}_n, f_{\boldsymbol{\theta}}(\boldsymbol{x}_n)) \in \mathbb{R}^{C \times C}$ as the Hessian of the loss w.r.t. the model outputs. Now we can write $\boldsymbol{s}_{\ell,n} = \boldsymbol{W}_\ell\boldsymbol{a}_{\ell,n} = (\boldsymbol{a}_{\ell,n}^\mathsf{T} \otimes \mathbf{I}_{P_{\ell,\text{out}}})\text{vec}(\boldsymbol{W}_\ell)$ and with this we have $\boldsymbol{J}_{\boldsymbol{\theta}_\ell}\boldsymbol{s}_{\ell,n} = \boldsymbol{a}_{\ell,n}^\mathsf{T} \otimes \mathbf{I}_{P_{\ell,\text{out}}}$. Additionally, by defining $\boldsymbol{b}_{\ell,n} := \boldsymbol{J}_{\boldsymbol{s}_{\ell,n}}f_{\boldsymbol{\theta}}(\boldsymbol{x}_n)^\mathsf{T} \in \mathbb{R}^{P_{\ell,\text{out}} \times C}$ as the transposed Jacobian of the model outputs w.r.t. to the pre-activations of the $\ell$-th layer, we note that $\boldsymbol{J}_{\boldsymbol{\theta}_\ell}(\boldsymbol{x}_n)^\mathsf{T} = (\boldsymbol{a}_{\ell,n}^\mathsf{T} \otimes \mathbf{I}_{P_{\ell,\text{out}}})^\mathsf{T}\boldsymbol{b}_{\ell,n} = \boldsymbol{a}_{\ell,n} \otimes \boldsymbol{b}_{\ell,n}$.

Replacing the (transposed) Jacobians in the definition of the GGN by this expression, we have

$$
\begin{aligned}
\mathbf{GGN}(\boldsymbol{\theta}_\ell) &= \sum_{n=1}^{N} \boldsymbol{J}_{\boldsymbol{\theta}_\ell}(\boldsymbol{x}_n)^\mathsf{T}\boldsymbol{\Lambda}(f_{\boldsymbol{\theta}}(\boldsymbol{x}_n))\boldsymbol{J}_{\boldsymbol{\theta}_\ell}(\boldsymbol{x}_n) \\
&= \sum_{n=1}^{N} (\boldsymbol{a}_{\ell,n} \otimes \boldsymbol{b}_{\ell,n})\boldsymbol{\Lambda}(f_{\boldsymbol{\theta}}(\boldsymbol{x}_n))(\boldsymbol{a}_{\ell,n} \otimes \boldsymbol{b}_{\ell,n})^\mathsf{T} \\
&= \sum_{n=1}^{N} (\boldsymbol{a}_{\ell,n}\boldsymbol{a}_{\ell,n}^\mathsf{T}) \otimes (\boldsymbol{b}_{\ell,n}\boldsymbol{\Lambda}(f_{\boldsymbol{\theta}}(\boldsymbol{x}_n))\boldsymbol{b}_{\ell,n}^\mathsf{T}) \\
&\approx \underbrace{\left[\frac{1}{N}\sum_{n=1}^{N}\boldsymbol{a}_{\ell,n}\boldsymbol{a}_{\ell,n}^\mathsf{T}\right]}_{=:\boldsymbol{A}_\ell} \otimes \underbrace{\left[\sum_{n=1}^{N}\boldsymbol{b}_{\ell,n}\boldsymbol{\Lambda}(f_{\boldsymbol{\theta}}(\boldsymbol{x}_n))\boldsymbol{b}_{\ell,n}^\mathsf{T}\right]}_{=:\boldsymbol{B}_\ell}.
\end{aligned}
\tag{21}
$$

This derivation is a bit more convenient for our purposes, as it does not require us to keep track of the expectation over the labels $\boldsymbol{y}$, while still being equivalent to the Fisher for common loss functions (c.f. Section 2.3). Moreover, in our context, it will be useful to have the Jacobians $\boldsymbol{J}_{\boldsymbol{\theta}_\ell}(\boldsymbol{x}_n)$ separate from the loss; therefore, we will only explicitly write our results for the GGN.

## B.2 K-FAC for linear weight-sharing layers

### B.2.1 The expand setting and K-FAC-expand

The expand setting can be identified by a loss with $N \cdot R$ terms, which corresponds to assuming $N \cdot R$ i.i.d. examples,

> **The Expand Setting**
>
> $$
> \mathcal{L}_{\text{expand}}(f_{\boldsymbol{\theta}}, \mathcal{D}) := -\sum_{n=1}^{N}\sum_{r=1}^{R} \log p(\boldsymbol{y}_{n,r}|f_{\boldsymbol{\theta}}(\boldsymbol{x}_n)_r),
> \tag{22}
> $$

where $f_{\boldsymbol{\theta}}(\boldsymbol{x}_n)_r$ is the $r$-th row of the model output $f_{\boldsymbol{\theta}}(\boldsymbol{x}_n) \in \mathbb{R}^{R \times C}$ and $\boldsymbol{y}_{n,r}$ is the $r$-th row of the label $\boldsymbol{Y}_n \in \mathbb{R}^{R \times C}$. A typical example of this type of loss function is language translation, where $N$ is the number of training examples and $R$ is the sequence length.

Note that we are not assuming our inputs $\boldsymbol{x}_n$ to have an additional weight-sharing dimension since we only require that the input to the $\ell$-th layer has this additional dimension, i.e. $\boldsymbol{A}_{\ell,n} \in \mathbb{R}^{R \times D}$. This obviously does not exclude the case where $\boldsymbol{x}_n$ already has this weight-sharing dimension, e.g. a sequence of tokens in translation tasks.

We can express the Jacobian of the $r$-th row of the model output $f_{\boldsymbol{\theta}}(\boldsymbol{x}_n) \in \mathbb{R}^{R \times C}$ w.r.t. the parameters $\boldsymbol{\theta}_\ell$ as

$$(\boldsymbol{J}_{\boldsymbol{\theta}_\ell}(\boldsymbol{x}_n)_r)_{ij} = \sum_{m=1}^{R} \sum_{p=1}^{P_{\ell,\text{out}}} \frac{\partial f_{\boldsymbol{\theta}}(\boldsymbol{x}_n)_{ri}}{\partial \boldsymbol{S}_{\ell,n,mp}} \frac{\partial \boldsymbol{S}_{\ell,n,mp}}{\partial \boldsymbol{\theta}_{\ell,j}} \tag{23}$$

or in matrix form

$$\boldsymbol{J}_{\boldsymbol{\theta}_\ell}(\boldsymbol{x}_n)_r = \sum_{m=1}^{R} \boldsymbol{J}_{\boldsymbol{s}_{\ell,n,m}} f_{\boldsymbol{\theta}}(\boldsymbol{x}_n)_r \boldsymbol{J}_{\boldsymbol{\theta}_\ell} \boldsymbol{s}_{\ell,n,m}. \tag{24}$$

Since the weights $\boldsymbol{\theta}_\ell$ are shared across the weight-sharing dimension of size $R$, we can write the $r$-th row of $\boldsymbol{S}_{\ell,n}$ as $\boldsymbol{s}_{\ell,n,r} = \boldsymbol{W}_\ell \boldsymbol{a}_{\ell,n,r}$ and we have $\boldsymbol{J}_{\boldsymbol{\theta}_\ell} \boldsymbol{s}_{\ell,n,r} = \boldsymbol{a}_{\ell,n,r}^{\mathsf{T}} \otimes \boldsymbol{I}_{P_{\ell,\text{out}}}$, as for regular K-FAC (c.f. Appendix B.1). We denote $\boldsymbol{b}_{\ell,n,r,k} := \boldsymbol{J}_{\boldsymbol{s}_{\ell,n,k}} f_{\boldsymbol{\theta}}(\boldsymbol{x}_n)_r^{\mathsf{T}}$. Hence, we have

$$\boldsymbol{J}_{\boldsymbol{\theta}_\ell}(\boldsymbol{x}_n)_r^{\mathsf{T}} = \left( \sum_{m=1}^{R} \boldsymbol{b}_{\ell,n,r,m}^{\mathsf{T}} (\boldsymbol{a}_{\ell,n,m}^{\mathsf{T}} \otimes \boldsymbol{I}_{P_{\ell,\text{out}}}) \right)^{\mathsf{T}}$$
$$= \sum_{m=1}^{R} \boldsymbol{a}_{\ell,n,m} \otimes \boldsymbol{b}_{\ell,n,r,m}. \tag{25}$$

On a high level, applying K-FAC to a model trained with this type of loss just requires treating the problem as if we had $N \cdot R$ *independent* examples and deriving the approximation in the same way as we would with $N$ examples (c.f. Appendix B.1),

$$\mathbf{GGN}(\boldsymbol{\theta}_\ell) = \sum_{n=1}^{N} \sum_{r=1}^{R} \boldsymbol{J}_{\boldsymbol{\theta}_\ell}(\boldsymbol{x}_n)_r^{\mathsf{T}} \boldsymbol{\Lambda}(f_{\boldsymbol{\theta}}(\boldsymbol{x}_n)_r) \boldsymbol{J}_{\boldsymbol{\theta}_\ell}(\boldsymbol{x}_n)_r$$
$$= \sum_{n=1}^{N} \sum_{r=1}^{R} \left( \sum_{m=1}^{R} \boldsymbol{a}_{\ell,n,m} \otimes \boldsymbol{b}_{\ell,n,r,m} \right) \boldsymbol{\Lambda}(f_{\boldsymbol{\theta}}(\boldsymbol{x}_n)_r) \left( \sum_{m'=1}^{R} \boldsymbol{a}_{\ell,n,m'}^{\mathsf{T}} \otimes \boldsymbol{b}_{\ell,n,r,m'}^{\mathsf{T}} \right). \tag{26}$$

However, we cannot directly write each of the $N \cdot R$ loss terms as a Kronecker product without any approximation. One approach could be to approximate each $\boldsymbol{J}_{\boldsymbol{\theta}_\ell}(\boldsymbol{x}_n)_r$ with a Kronecker product of sums, as the K-FAC approximation does for the sum of the $N$ data points, but then we would have to be able to access $\sum_{m=1}^{R} \boldsymbol{b}_{\ell,n,r,m}$. Moreover, this would not even be exact in the simple settings we consider later. In contrast, what can be implemented in practice without additional backward passes (c.f. Appendix B.4) and what has been used for convolutional neural networks (Grosse & Martens, 2016) and language transformers (Grosse et al., 2023; Osawa et al., 2022; Pauloski et al., 2021; Zhang et al., 2019a) is

$$\mathbf{GGN}(\boldsymbol{\theta}_\ell) \approx \sum_{n=1}^{N} \sum_{r=1}^{R} \sum_{m=1}^{R} (\boldsymbol{a}_{\ell,n,m} \otimes \boldsymbol{b}_{\ell,n,r,m}) \boldsymbol{\Lambda}(f_{\boldsymbol{\theta}}(\boldsymbol{x}_n)_r) (\boldsymbol{a}_{\ell,n,m}^{\mathsf{T}} \otimes \boldsymbol{b}_{\ell,n,r,m}^{\mathsf{T}})$$
$$= \sum_{n=1}^{N} \sum_{m=1}^{R} (\boldsymbol{a}_{\ell,n,m} \boldsymbol{a}_{\ell,n,m}^{\mathsf{T}}) \otimes \left( \sum_{r=1}^{R} \boldsymbol{b}_{\ell,n,r,m} \boldsymbol{\Lambda}(f_{\boldsymbol{\theta}}(\boldsymbol{x}_n)_r) \boldsymbol{b}_{\ell,n,r,m}^{\mathsf{T}} \right), \tag{27}$$

where we ignore all the terms with $m \neq m'$, which allows us to express each of the $N \cdot R$ terms as a Kronecker product.[7] Consequently, we can apply the regular K-FAC approximation over $N \cdot R$ terms instead of just $N$ terms as usual. We call the resulting approximation *K-FAC-expand*:

---

[7]However, the authors of the work on transformers do not discuss this extension of K-FAC at all; although Zhang et al. (2018) do explicitly discuss the tying of the embedding and linear output layer weights.

> **K-FAC-expand**
>
> $$\mathbf{G\hat{G}N}_{\boldsymbol{\theta}_\ell}^{\text{expand}} := \underbrace{\left[\frac{1}{NR}\sum_{n=1}^{N}\sum_{m=1}^{R}\boldsymbol{a}_{\ell,n,m}\boldsymbol{a}_{\ell,n,m}^{\mathsf{T}}\right]}_{=\boldsymbol{A}_\ell} \otimes \underbrace{\left[\sum_{n=1}^{N}\sum_{m=1}^{R}\sum_{r=1}^{R}\boldsymbol{b}_{\ell,n,r,m}\boldsymbol{\Lambda}(f_{\boldsymbol{\theta}}(\boldsymbol{x}_n)_r)\boldsymbol{b}_{\ell,n,r,m}^{\mathsf{T}}\right]}_{=\boldsymbol{B}_\ell}$$
>
> $$(28)$$

There is one simple case where the exact expression in Equation (26) is identical to the approximation in Equation (27). When $\boldsymbol{b}_{\ell,n,r,m} = \boldsymbol{0}$ for all $r \neq m$, both expression are equivalent to

$$\sum_{n=1}^{N}\sum_{r=1}^{R}\left(\boldsymbol{a}_{\ell,n,r}\otimes\boldsymbol{b}_{\ell,n,r,r}\right)\boldsymbol{\Lambda}(f_{\boldsymbol{\theta}}(\boldsymbol{x}_n)_r)\left(\boldsymbol{a}_{\ell,n,r}^{\mathsf{T}}\otimes\boldsymbol{b}_{\ell,n,r,r}^{\mathsf{T}}\right). \tag{29}$$

With other words, when $f_{\boldsymbol{\theta}}(\boldsymbol{x}_n)_r$ is independent of all pre-activations $\boldsymbol{s}_{\ell,n,m}$ with $m \neq r$ the two expressions coincide. This does not even hold for simplistic transformer models since the self-attention mechanism (Section 2.2) in transformers directly correlates elements across the weight-sharing dimension – we discuss this in more detail in Appendix B.3.1. Alternatively, we could also scale Equation (27) by $R$, which leads to a scaling of $1/N$ instead of $1/NR$ in Equation (28). This might be a better approximation when $\boldsymbol{b}_{\ell,n,r,m} \neq \boldsymbol{0}$ for $r \neq m$ and is also what has been used by previous work on transformers (Grosse et al., 2023; Osawa et al., 2022; Pauloski et al., 2021; Zhang et al., 2019a). However, we choose to use the scaling in Equation (28) since the condition above holds for networks that simply stack multiple linear weight-sharing layers; this allows us to show that the approximation in Equation (28) is exact in the same simple cases as regular K-FAC.

For a typical neural network with nonlinear activation functions, K-FAC is only an approximation. However, for regular individual linear layers and deep linear networks, K-FAC is known to be exact assuming a Gaussian likelihood (Bernacchia et al., 2018). While this holds for the full GGN/Fisher, we only focus on the block-diagonal case here. To motivate K-FAC-expand, we want to show that similar statements hold for a single linear weight-sharing layer and deep linear networks with weight-sharing in the expand setting. First, we state a simple condition for which the approximation is indeed exact; this line of reasoning could also be applied to K-FAC for regular linear layers since only the effective number of data points changes from $N \cdot R$ to $N$. Note, that we could also state more trivial sufficient conditions for the exactness of the approximation, i.e. $N = R = 1$ and when all inputs to a layer $\boldsymbol{a}_{\ell,n,r}$ are the same for all $n \in \{1,\ldots,N\}$ and $r \in \{1,\ldots,R\}$. We do not state these types of conditions explicitly from now on.

**Lemma 3 (Sufficient condition for exactness of K-FAC-expand in the expand setting).** *Let $\boldsymbol{C}_\ell \in \mathbb{R}^{P_{\ell,\text{out}} \times P_{\ell,\text{out}}}$ be a constant matrix for layer $\ell$. If $\boldsymbol{b}_{\ell,n,r,m} = \boldsymbol{0}$ for all $r \neq m$ and $\boldsymbol{b}_{\ell,n,m,m}\boldsymbol{\Lambda}(f_{\boldsymbol{\theta}}(\boldsymbol{x}_n)_m)\boldsymbol{b}_{\ell,n,m,m}^{\mathsf{T}} = \boldsymbol{C}_\ell$ for all $n \in \{1,\ldots,N\}$ and $m \in \{1,\ldots,R\}$, then the K-FAC approximation in Equation (7) is equal to the exact GGN/Fisher of the $\ell$-th layer in the expand setting.*

*Proof.* As mentioned before, when $\boldsymbol{b}_{\ell,n,r,m} = \boldsymbol{0}$ for all $r \neq m$ the last line of Equation (26) and the first line of Equation (27) both simplify to Equation (29). Hence, we can directly show that the second and third approximation in Equation (27) equal the exact expression for the GGN of layer $\ell$ from there. We have

$$\left(\frac{1}{NR}\sum_{n=1}^{N}\sum_{r=1}^{R}\boldsymbol{a}_{\ell,n,r}\boldsymbol{a}_{\ell,n,r}^{\mathsf{T}}\right) \otimes \left(\sum_{n=1}^{N}\sum_{r=1}^{R}\boldsymbol{b}_{\ell,n,r,r}\boldsymbol{\Lambda}(f_{\boldsymbol{\theta}}(\boldsymbol{x}_n)_r)\boldsymbol{b}_{\ell,n,r,r}^{\mathsf{T}}\right)$$

$$= \left(\frac{1}{NR}\sum_{n=1}^{N}\sum_{r=1}^{R}\boldsymbol{a}_{\ell,n,r}\boldsymbol{a}_{\ell,n,r}^{\mathsf{T}}\right) \otimes (NR\boldsymbol{C}_\ell)$$

$$= \left(\sum_{n=1}^{N}\sum_{r=1}^{R}\boldsymbol{a}_{\ell,n,r}\boldsymbol{a}_{\ell,n,r}^{\mathsf{T}}\right) \otimes \boldsymbol{C}_\ell \tag{30}$$

$$= \sum_{n=1}^{N}\sum_{r=1}^{R}\left(\boldsymbol{a}_{\ell,n,r}\boldsymbol{a}_{\ell,n,r}^{\mathsf{T}}\right) \otimes \left(\boldsymbol{b}_{\ell,n,r,r}\boldsymbol{\Lambda}(f_{\boldsymbol{\theta}}(\boldsymbol{x}_n)_r)\boldsymbol{b}_{\ell,n,r,r}^{\mathsf{T}}\right),$$

where we have used the assumption that $\boldsymbol{b}_{\ell,n,m,m}\boldsymbol{\Lambda}(f_{\boldsymbol{\theta}}(\boldsymbol{x}_n)_m)\boldsymbol{b}_{\ell,n,m,m}^{\mathsf{T}} = \boldsymbol{C}_\ell$ is the same for all $n \in \{1, \ldots, N\}$ and $m \in \{1, \ldots, R\}$. $\hfill\square$

Leveraging this simple insight, we can provide an example of a single layer where the assumptions of Lemma 3 are fulfilled.

**Proposition 4 (Exactness of K-FAC-expand for single linear layer in the expand setting).** *For a single linear weight-sharing layer and a Gaussian likelihood with p.d. covariance matrix $\boldsymbol{\Sigma} \in \mathbb{R}^{C\times C}$, K-FAC-expand is exact in the expand setting.*

*Proof.* We can write $f_{\boldsymbol{\theta}}(\boldsymbol{x}_n)_r = \boldsymbol{W}_\ell\boldsymbol{x}_{n,r}$ and hence $\boldsymbol{b}_{\ell,n,r,m} = \boldsymbol{0}$ for $r \neq m$. Moreover, we have $\boldsymbol{\Lambda}(f_{\boldsymbol{\theta}}(\boldsymbol{x}_n)_r) = \boldsymbol{\Sigma}^{-1}$ and $\boldsymbol{b}_{\ell,n,m,m} = \mathbf{I}_C$ ($P_{\ell,\mathrm{out}} = C$ for a single layer). Hence,

$$\boldsymbol{b}_{\ell,n,m,m}\boldsymbol{\Lambda}(f_{\boldsymbol{\theta}}(\boldsymbol{x}_n)_r)\boldsymbol{b}_{\ell,n,m,m}^{\mathsf{T}} = \mathbf{I}_C\boldsymbol{\Sigma}^{-1}\mathbf{I}_C = \boldsymbol{\Sigma}^{-1}$$

for all $n \in \{1, \ldots, N\}$ and $m \in \{1, \ldots, R\}$. Therefore, the desired result follows from Lemma 3. $\hfill\square$

A natural question might be if the same result also holds for *deep* linear networks. A deep linear network is here defined as a model of the form

$$f_{\boldsymbol{\theta}}(\boldsymbol{x}) = \boldsymbol{W}_L \ldots \boldsymbol{W}_\ell \ldots \boldsymbol{W}_1\boldsymbol{x} = \boldsymbol{W}\boldsymbol{x}, \tag{31}$$

where $\boldsymbol{x} \in \mathbb{R}^D$ and $\boldsymbol{W}_L \in \mathbb{R}^{C\times P_{L,\mathrm{in}}}, \boldsymbol{W}_\ell \in \mathbb{R}^{P_{\ell,\mathrm{out}}\times P_{\ell,\mathrm{in}}}$ (with $P_{\ell,\mathrm{in}} = P_{\ell-1,\mathrm{out}}$), and $\boldsymbol{W}_1 \in \mathbb{R}^{P_{1,\mathrm{out}}\times D}$. Decomposing a single weight matrix $\boldsymbol{W}$ into $L$ separate ones is a common framework for theoretical analysis since it creates nonlinear training dynamics for gradient-based training algorithms, while still having analytical solutions (Bernacchia et al., 2018; Saxe et al., 2014). We adopt the notation of Bernacchia et al. (2018) and define

$$\boldsymbol{W}_\ell^a := \boldsymbol{W}_L \ldots \boldsymbol{W}_{\ell+1} \tag{32}$$

as the product of the weight matrices *ahead* of $\boldsymbol{W}_\ell$ and

$$\boldsymbol{W}_\ell^b := \boldsymbol{W}_{\ell-1} \ldots \boldsymbol{W}_1 \tag{33}$$

as the product of the weight matrices *behind* of $\boldsymbol{W}_\ell$. Hence, we can write $f_{\boldsymbol{\theta}}(\boldsymbol{x}) = \boldsymbol{W}_\ell^a\boldsymbol{W}_\ell\boldsymbol{W}_\ell^b\boldsymbol{x}$. Note, that now

$$\boldsymbol{a}_{\ell,n,r} = \boldsymbol{W}_\ell^b\boldsymbol{x}_{n,r} \in \mathbb{R}^{P_{\ell,\mathrm{in}}} \tag{34}$$

and

$$\boldsymbol{b}_{\ell,n,r,r} = \boldsymbol{W}_\ell^{a^{\mathsf{T}}} \in \mathbb{R}^{P_{\ell,\mathrm{out}}\times C}. \tag{35}$$

Using these insights, we can now easily state the result for deep linear networks.

**Proposition 1 (Exactness of K-FAC-expand for deep linear network in the expand setting).** *For layer $\ell$ of a deep linear network defined as in Equation (8) and a Gaussian likelihood with p.d. covariance matrix $\boldsymbol{\Sigma} \in \mathbb{R}^{C\times C}$, K-FAC-expand is exact in the expand setting.*

*Proof.* We can write $f_{\boldsymbol{\theta}}(\boldsymbol{x}_n)_r = \boldsymbol{W}\boldsymbol{x}_{n,r}$ and hence $\boldsymbol{b}_{\ell,n,r,m} = \boldsymbol{0}$ for $r \neq m$. We have $\boldsymbol{\Lambda}(f_{\boldsymbol{\theta}}(\boldsymbol{x}_n)_r) = \boldsymbol{\Sigma}^{-1}$ and $\boldsymbol{b}_{\ell,n,r,r} = \boldsymbol{W}_\ell^{a^{\mathsf{T}}}$. Hence, $\boldsymbol{b}_{\ell,n,m,m}\boldsymbol{\Lambda}(f_{\boldsymbol{\theta}}(\boldsymbol{x}_n)_m)\boldsymbol{b}_{\ell,n,m,m}^{\mathsf{T}} = \boldsymbol{W}_\ell^{a^{\mathsf{T}}}\boldsymbol{\Sigma}^{-1}\boldsymbol{W}_\ell^a$ for all $n \in \{1, \ldots, N\}$ and $m \in \{1, \ldots, R\}$. Therefore, the desired result follows from Lemma 3. $\hfill\square$

### B.2.2 The reduce setting and K-FAC-reduce

The reduce setting is characterized by a loss with just $N$ loss terms, i.e.

**The Reduce Setting**

$$\mathcal{L}_{\mathrm{reduce}}(f_{\boldsymbol{\theta}}, \mathcal{D}) := -\sum_{n=1}^{N}\log p(\boldsymbol{y}_n|f_{\boldsymbol{\theta}}(\boldsymbol{x}_n)), \tag{36}$$

where the crucial observation is that the weight-sharing dimension must have been reduced somewhere in the forward pass of the neural network $f_{\boldsymbol{\theta}}$. A typical instance where this type of loss is used together

with a model with linear weight-sharing layers is image classification with a vision transformer or a convolutional neural network. Note, that the inputs $\boldsymbol{x}_n$ and labels $\boldsymbol{y}_n$ do not have a weight-sharing dimension here; in general, it is also possible for the inputs to have this additional dimension of size $R$ already.

Since $\boldsymbol{A}_{\ell,n} \in \mathbb{R}^{R \times P_{\ell,\mathrm{in}}}$ is now a matrix, we have $\boldsymbol{S}_{\ell,n} = \boldsymbol{A}_{\ell,n} \boldsymbol{W}_\ell^\mathsf{T} \in \mathbb{R}^{R \times P_{\ell,\mathrm{out}}}$. Hence, $\boldsymbol{J}_{\boldsymbol{\theta}_\ell} \boldsymbol{S}_{\ell,n}$ and $\boldsymbol{J}_{\boldsymbol{S}_{\ell,n}} f_{\boldsymbol{\theta}}(\boldsymbol{x}_n)$ are now both multi-dimensional arrays. Luckily, we can simplify this by writing

$$(\boldsymbol{J}_{\boldsymbol{\theta}_\ell} f_{\boldsymbol{\theta}}(\boldsymbol{x}_n))_{ij} = \sum_{r=1}^{R} \sum_{p=1}^{P_{\ell,\mathrm{out}}} \frac{\partial f_{\boldsymbol{\theta}}(\boldsymbol{x}_n)_i}{\partial \boldsymbol{S}_{\ell,n,rp}} \frac{\partial \boldsymbol{S}_{\ell,n,rp}}{\partial \boldsymbol{\theta}_{\ell,j}}, \tag{37}$$

or in matrix form

$$\boldsymbol{J}_{\boldsymbol{\theta}_\ell} f_{\boldsymbol{\theta}}(\boldsymbol{x}_n) = \sum_{r=1}^{R} \boldsymbol{J}_{\boldsymbol{s}_{\ell,n,r}} f_{\boldsymbol{\theta}}(\boldsymbol{x}_n) \boldsymbol{J}_{\boldsymbol{\theta}_\ell} \boldsymbol{s}_{\ell,n,r}, \tag{38}$$

where $\boldsymbol{s}_{\ell,n,r} \in \mathbb{R}^{P_{\ell,\mathrm{out}}}$ is the $r$-th row of $\boldsymbol{S}_{\ell,n}$ and $\boldsymbol{s}_{\ell,n,r} = \boldsymbol{W}_\ell \boldsymbol{a}_{\ell,n,r}$.

Using this equivalence we can approximate the GGN for layer $\ell$ as

$$
\begin{aligned}
\mathbf{GGN}(\boldsymbol{\theta}_\ell) &= \sum_{n=1}^{N} \boldsymbol{J}_{\boldsymbol{\theta}_\ell}(\boldsymbol{x}_n)^\mathsf{T} \boldsymbol{\Lambda}(f_{\boldsymbol{\theta}}(\boldsymbol{x}_n)) \boldsymbol{J}_{\boldsymbol{\theta}_\ell}(\boldsymbol{x}_n) \\
&= \sum_{n=1}^{N} \left( \sum_{r=1}^{R} \boldsymbol{J}_{\boldsymbol{s}_{\ell,n,r}} f_{\boldsymbol{\theta}}(\boldsymbol{x}_n) \boldsymbol{J}_{\boldsymbol{\theta}_\ell} \boldsymbol{s}_{\ell,n,r} \right)^\mathsf{T} \boldsymbol{\Lambda}(f_{\boldsymbol{\theta}}(\boldsymbol{x}_n)) \left( \sum_{r=1}^{R} \boldsymbol{J}_{\boldsymbol{s}_{\ell,n,r}} f_{\boldsymbol{\theta}}(\boldsymbol{x}_n) \boldsymbol{J}_{\boldsymbol{\theta}_\ell} \boldsymbol{s}_{\ell,n,r} \right) \\
&= \sum_{n=1}^{N} \left( \sum_{r=1}^{R} \boldsymbol{a}_{\ell,n,r} \otimes \boldsymbol{b}_{\ell,n,r} \right) \boldsymbol{\Lambda}(f_{\boldsymbol{\theta}}(\boldsymbol{x}_n)) \left( \sum_{r=1}^{R} \boldsymbol{a}_{\ell,n,r} \otimes \boldsymbol{b}_{\ell,n,r} \right)^\mathsf{T} \\
&\approx \sum_{n=1}^{N} \underbrace{\left[ \frac{1}{R} \sum_{r=1}^{R} \boldsymbol{a}_{\ell,n,r} \right]}_{=: \hat{\boldsymbol{a}}_{\ell,n}} \otimes \underbrace{\left[ \sum_{r=1}^{R} \boldsymbol{b}_{\ell,n,r} \right]}_{=: \hat{\boldsymbol{b}}_{\ell,n}} \boldsymbol{\Lambda}(f_{\boldsymbol{\theta}}(\boldsymbol{x}_n)) \left[ \frac{1}{R} \sum_{r=1}^{R} \boldsymbol{a}_{\ell,n,r}^\mathsf{T} \right] \otimes \left[ \sum_{r=1}^{R} \boldsymbol{b}_{\ell,n,r}^\mathsf{T} \right] \\
&= \sum_{n=1}^{N} \left( \hat{\boldsymbol{a}}_{\ell,n} \hat{\boldsymbol{a}}_{\ell,n}^\mathsf{T} \right) \otimes \left( \hat{\boldsymbol{b}}_{\ell,n} \boldsymbol{\Lambda}(f_{\boldsymbol{\theta}}(\boldsymbol{x}_n)) \hat{\boldsymbol{b}}_{\ell,n}^\mathsf{T} \right) \\
&\approx \underbrace{\left[ \frac{1}{N} \sum_{n=1}^{N} \hat{\boldsymbol{a}}_{\ell,n} \hat{\boldsymbol{a}}_{\ell,n}^\mathsf{T} \right]}_{=: \hat{\boldsymbol{A}}_\ell} \otimes \underbrace{\left[ \sum_{n=1}^{N} \hat{\boldsymbol{b}}_{\ell,n} \boldsymbol{\Lambda}(f_{\boldsymbol{\theta}}(\boldsymbol{x}_n)) \hat{\boldsymbol{b}}_{\ell,n}^\mathsf{T} \right]}_{=: \hat{\boldsymbol{B}}_\ell},
\end{aligned}
\tag{39}
$$

where we have approximated the sum over the $R$ Kronecker products with a Kronecker product of sums for each of the $N$ per-input Jacobians, before applying the same type of approximation as usual to the sum over the $N$ data points. This approximation has been proposed in Tang et al. (2021) to improve the efficiency of their proposed K-FAC variation (SKFAC) for convolutions by reducing the spatial dimension, purely based on the empirical observation that it works well in practice. The idea to approximate the Jacobians within the GGN with a Kronecker-product has also been proposed in the context of invariance learning with deep neural networks via differentiable Laplace approximations in Immer et al. (2022). We call the approximation in Equation (39) *K-FAC-reduce* and to highlight the difference to K-FAC-expand, we can rewrite it as

$$\mathbf{G\hat{G}N}_{\boldsymbol{\theta}_\ell}^{\text{reduce}} :=$$

$$\underbrace{\left[ \frac{1}{NR^2} \sum_{n=1}^{N} \left( \sum_{r=1}^{R} \boldsymbol{a}_{\ell,n,r} \right) \left( \sum_{r=1}^{R} \boldsymbol{a}_{\ell,n,r}^{\mathsf{T}} \right) \right]}_{=\hat{\boldsymbol{A}}_\ell} \otimes \underbrace{\left[ \sum_{n=1}^{N} \left( \sum_{r=1}^{R} \boldsymbol{b}_{\ell,n,r} \right) \boldsymbol{\Lambda}(f_{\boldsymbol{\theta}}(\boldsymbol{X}_n)) \left( \sum_{r=1}^{R} \boldsymbol{b}_{\ell,n,r}^{\mathsf{T}} \right) \right]}_{=\hat{\boldsymbol{B}}_\ell}.$$

$$(40)$$

As for K-FAC-expand, we want to show that this approximation can be exact in the case of a single layer or a deep linear network and a Gaussian likelihood. First, we state an analogous condition to Lemma 3.

**Lemma 5** (**Sufficient condition for exactness of K-FAC-reduce in the reduce setting**). *Let* $\boldsymbol{D}_{\ell,n} \in \mathbb{R}^{P_{\ell,\text{out}} \times C}$ *be a constant matrix for layer* $\ell$ *and data point* $\boldsymbol{x}_n$. *Further, let* $\boldsymbol{C}_\ell \in \mathbb{R}^{P_{\ell,\text{out}} \times P_{\ell,\text{out}}}$ *be a constant matrix for layer* $\ell$. *If it holds for each* $n$ *that* $\boldsymbol{b}_{\ell,n,r} = \boldsymbol{D}_{\ell,n}$ *for all* $r \in \{1, \ldots, R\}$ *and* $\hat{\boldsymbol{b}}_{\ell,n} \boldsymbol{\Lambda}(f_{\boldsymbol{\theta}}(\boldsymbol{x}_n)) \hat{\boldsymbol{b}}_{\ell,n}^{\mathsf{T}} = \boldsymbol{C}_\ell$ *for all* $n \in \{1, \ldots, N\}$, *then the K-FAC-reduce approximation in Equation* (10) *is equal to the exact GGN of the* $\ell$-*th layer in the reduce setting.*

*Proof.* We start with the first approximation and derive the exactness of this step under our assumptions. We have

$$\sum_{n=1}^{N} \left( \frac{1}{R} \sum_{r=1}^{R} \boldsymbol{a}_{\ell,n,r} \otimes \sum_{r=1}^{R} \boldsymbol{b}_{\ell,n,r} \right) \boldsymbol{\Lambda}(f_{\boldsymbol{\theta}}(\boldsymbol{x}_n)) \left( \frac{1}{R} \sum_{r=1}^{R} \boldsymbol{a}_{\ell,n,r} \otimes \sum_{r=1}^{R} \boldsymbol{b}_{\ell,n,r} \right)^{\mathsf{T}}$$

$$= \sum_{n=1}^{N} \left( \frac{1}{R} \sum_{r=1}^{R} \boldsymbol{a}_{\ell,n,r} \otimes R\boldsymbol{D}_{\ell,n} \right) \boldsymbol{\Lambda}(f_{\boldsymbol{\theta}}(\boldsymbol{x}_n)) \left( \frac{1}{R} \sum_{r=1}^{R} \boldsymbol{a}_{\ell,n,r} \otimes R\boldsymbol{D}_{\ell,n} \right)^{\mathsf{T}} \qquad (41)$$

$$= \sum_{n=1}^{N} \left( \sum_{r=1}^{R} \boldsymbol{a}_{\ell,n,r} \otimes \boldsymbol{b}_{\ell,n,r} \right) \boldsymbol{\Lambda}(f_{\boldsymbol{\theta}}(\boldsymbol{x}_n)) \left( \sum_{r=1}^{R} \boldsymbol{a}_{\ell,n,r} \otimes \boldsymbol{b}_{\ell,n,r} \right)^{\mathsf{T}},$$

where we have used the assumption that for each $n$, we have $\boldsymbol{b}_{\ell,n,r} = \boldsymbol{D}_{\ell,n}$ for all $r \in \{1, \ldots, R\}$. Now we consider the second approximation in Equation (39). Analogously, we have

$$\left( \frac{1}{N} \sum_{n=1}^{N} \hat{\boldsymbol{a}}_{\ell,n} \hat{\boldsymbol{a}}_{\ell,n}^{\mathsf{T}} \right) \otimes \left( \sum_{n=1}^{N} \hat{\boldsymbol{b}}_{\ell,n} \boldsymbol{\Lambda}(f_{\boldsymbol{\theta}}(\boldsymbol{x}_n)) \hat{\boldsymbol{b}}_{\ell,n}^{\mathsf{T}} \right)$$

$$= \left( \frac{1}{N} \sum_{n=1}^{N} \hat{\boldsymbol{a}}_{\ell,n} \hat{\boldsymbol{a}}_{\ell,n}^{\mathsf{T}} \right) \otimes N\boldsymbol{C}_\ell \qquad (42)$$

$$= \sum_{n=1}^{N} \left( \hat{\boldsymbol{a}}_{\ell,n} \hat{\boldsymbol{a}}_{\ell,n}^{\mathsf{T}} \right) \otimes \left( \hat{\boldsymbol{b}}_{\ell,n} \boldsymbol{\Lambda}(f_{\boldsymbol{\theta}}(\boldsymbol{x}_n)) \hat{\boldsymbol{b}}_{\ell,n}^{\mathsf{T}} \right),$$

where we have used that $\hat{\boldsymbol{b}}_{\ell,n} \boldsymbol{\Lambda}(f_{\boldsymbol{\theta}}(\boldsymbol{x}_n)) \hat{\boldsymbol{b}}_{\ell,n}^{\mathsf{T}} = \boldsymbol{C}_\ell$ for all $n \in \{1, \ldots, N\}$. $\square$

Until now, we did not have to explicitly take the aggregation function $z : \mathbb{R}^{R \times P_{\ell,\text{out}}} \to \mathbb{R}^{P_{\ell,\text{out}}}$ into account, since its Jacobian is simply subsumed in $\boldsymbol{b}_{\ell,n,r}$. Since we want to verify that the approximation in the reduce case is also exact in the simple scenarios from Proposition 1 and Proposition 2, we now have to also check if the Jacobian $\boldsymbol{J}_{\boldsymbol{s}_{\ell,n,r}} z_{\ell,n}$ with $\boldsymbol{z}_{\ell,n} := z(\boldsymbol{S}_{\ell,n}) \in \mathbb{R}^{P_{\ell,\text{out}}}$ is the same for all $r \in \{1, \ldots, R\}$, to make sure the first condition in Lemma 5 is fulfilled. Maybe

the simplest case where this holds is a scaled sum, i.e.

$$
\begin{aligned}
z(\boldsymbol{S}_{\ell,n}) &= c \sum_{r=1}^{R} \boldsymbol{s}_{\ell,n,r} \\
&= c \boldsymbol{S}_{\ell,n}^{\mathsf{T}} \boldsymbol{1}_R \\
&= \left( \boldsymbol{1}_R^{\mathsf{T}} \otimes c \mathbf{I}_{P_{\ell,\mathrm{out}}} \right) \mathrm{vec}(\boldsymbol{S}_{\ell,n}^{\mathsf{T}}) \\
&= \left( \boldsymbol{1}_R^{\mathsf{T}} \otimes c \mathbf{I}_{P_{\ell,\mathrm{out}}} \right) \boldsymbol{K}^{(R,P_{\ell,\mathrm{out}})} \mathrm{vec}(\boldsymbol{S}_{\ell,n})
\end{aligned}
\tag{43}
$$

with $c \in \mathbb{R}$ and the commutation matrix

$$
\boldsymbol{K}^{(R,P_{\ell,\mathrm{out}})} := \sum_{r=1}^{R} \sum_{p=1}^{P_{\ell,\mathrm{out}}} (\boldsymbol{e}_{R,r} \boldsymbol{e}_{P_{\ell,\mathrm{out}},p}^{\mathsf{T}}) \otimes (\boldsymbol{e}_{P_{\ell,\mathrm{out}},p} \boldsymbol{e}_{R,r}^{\mathsf{T}}),
\tag{44}
$$

where $\boldsymbol{e}_{i,j}$ is the $j$-th canonical vector of dimension $i$. This is a linear function in $\mathrm{vec}(\boldsymbol{S}_{\ell,n})$ and we have $\boldsymbol{J}_{\boldsymbol{s}_{\ell,n,r}} \boldsymbol{z}_{\ell,n} = c \mathbf{I}_{P_{\ell,\mathrm{out}}}$ for all $r \in \{1,\ldots,R\}$. In particular, when $c = 1$ the aggregation function is a simple sum and when $c = 1/R$ it is the mean. Notably, it is *not* sufficient for $z$ to be linear in $\mathrm{vec}(\boldsymbol{S}_{\ell,n})$, because as soon as we have a weighted sum with weights $c_r \in \mathbb{R}$ and they are not the same for all $r \in \{1,\ldots,R\}$, the Jacobians $\boldsymbol{J}_{\boldsymbol{s}_{\ell,n,r}} \boldsymbol{z}_{\ell,n}$ will also not be the same anymore. Both vision transformers and convolutional neural networks with average pooling use scaled sums as the aggregation function (with $c = 1/R$).

After clarifying the role of the aggregation function in the exactness of K-FAC-reduce, we can now state a similar statement to Proposition 4.

**Proposition 6** (**Exactness of K-FAC-reduce for single linear layer in the reduce setting**). *For a single linear layer, a Gaussian likelihood with p.d. covariance matrix $\boldsymbol{\Sigma} \in \mathbb{R}^{C \times C}$, and a scaled sum as defined in Equation (43) as the aggregation function applied to the output of the linear function, K-FAC-reduce is exact in the reduce setting.*

*Proof.* We have $\boldsymbol{\Lambda}(f_{\boldsymbol{\theta}}(\boldsymbol{x}_n)) = \boldsymbol{\Sigma}^{-1}$ and $\boldsymbol{b}_{\ell,n,r} = \left( \boldsymbol{J}_{\boldsymbol{z}_{\ell,n}} f_{\boldsymbol{\theta}}(\boldsymbol{x}_n) \boldsymbol{J}_{\boldsymbol{s}_{\ell,n,r}} \boldsymbol{z}_{\ell,n} \right)^{\mathsf{T}} = c \mathbf{I}_C$ for all $r \in \{1,\ldots,R\}$ and $n \in \{1,\ldots,N\}$ ($P_{\ell,\mathrm{out}} = C$ for a single layer). Hence, $\hat{\boldsymbol{b}}_{\ell,n} \boldsymbol{\Lambda}(f_{\boldsymbol{\theta}}(\boldsymbol{x}_n)) \hat{\boldsymbol{b}}_{\ell,n}^{\mathsf{T}} = c^2 R^2 \mathbf{I}_C \boldsymbol{\Sigma}^{-1} \mathbf{I}_C = c^2 R^2 \boldsymbol{\Sigma}^{-1}$ for all $n \in \{1,\ldots,N\}$. Therefore, the desired result follows from Lemma 5. $\square$

Just as for K-FAC-expand, we can extend this result to deep linear networks.

**Proposition 2** (**Exactness of K-FAC-reduce for deep linear network in the reduce setting**). *For layer $\ell$ of a deep linear network (Equation (8)), a Gaussian likelihood with p.d. covariance matrix $\boldsymbol{\Sigma} \in \mathbb{R}^{C \times C}$, and a scaled sum aggregation function, K-FAC-reduce is exact in the reduce setting.*

*Proof.* We have $\boldsymbol{\Lambda}(f_{\boldsymbol{\theta}}(\boldsymbol{x}_n)_r) = \boldsymbol{\Sigma}^{-1}$ and $\boldsymbol{b}_{\ell,n,r} = \left( \boldsymbol{J}_{\boldsymbol{z}_{\ell,n}} f_{\boldsymbol{\theta}}(\boldsymbol{x}_n) \boldsymbol{J}_{\boldsymbol{s}_{\ell,n,r}} \boldsymbol{z}_{\ell,n} \right)^{\mathsf{T}} = c \boldsymbol{W}_{\ell}^{a\mathsf{T}}$ for all $r \in \{1,\ldots,R\}$ and $n \in \{1,\ldots,N\}$. Hence,

$$
\hat{\boldsymbol{b}}_{\ell,n} \boldsymbol{\Lambda}(f_{\boldsymbol{\theta}}(\boldsymbol{x}_n)) \hat{\boldsymbol{b}}_{\ell,n}^{\mathsf{T}} = c^2 R^2 \boldsymbol{W}_{\ell}^{a\mathsf{T}} \boldsymbol{\Sigma}^{-1} \boldsymbol{W}_{\ell}^{a}
$$

for all $n \in \{1,\ldots,N\}$. Therefore, the desired result follows from Lemma 5. $\square$

To summarise, the difference between the expand and the reduce setting is at what point the aggregation over the additional weight-sharing dimension happens. If this dimension is not aggregated before the per-example loss, i.e. if the loss can be expanded to $N \cdot R$ instead of $N$ terms, we call it the expand setting. If the aggregation happens inside the model, we call it the reduce setting. Both settings motivate an approximation each, K-FAC-expand and K-FAC-reduce. Moreover, we presented simple cases where the approximations are exact. In Figure 1 and Figure 2 we verify this numerically, and also show that using the inappropriate approximation results in an inexact computation.

**Remark 1.** *In practice, however, both approximations can be applied in each of the two settings.*

## B.3 Examples of K-FAC for linear weight-sharing layers in the wild

### B.3.1 K-FAC for self-attention

While we have mentioned (vision) transformers for translation and image classification as prototypical examples for the expand and the reduce setting, we have mostly ignored how linear weight-sharing layers are used within the architecture and how this affects the approximation quality of K-FAC-expand and K-FAC-reduce. Linear weight-sharing layers are crucial for the self-attention mechanism in Section 2.2. To gain some intuition for models using this type of attention mechanism, we look at a network that only consists of one simplified variation of the self-attention mechanism used in transformers (we ignore the softmax function, but also consider a linear projection of the output with weight matrix $\boldsymbol{W}^V$), i.e.

$$f_{\boldsymbol{\theta}}(\boldsymbol{X}_n) = \underbrace{\boldsymbol{X}_n \boldsymbol{W}^{Q^\mathsf{T}}}_{=:\boldsymbol{S}_{Q,n}} \underbrace{\boldsymbol{W}^K \boldsymbol{X}_n^\mathsf{T}}_{=:\boldsymbol{S}_{K,n}^\mathsf{T}} \underbrace{\boldsymbol{X}_n \boldsymbol{W}^{V^\mathsf{T}}}_{=:\boldsymbol{S}_{V,n}}. \tag{45}$$

We can observe that it is no longer a linear function in the input $\boldsymbol{X}_n \in \mathbb{R}^{R \times D}$ and that we have three linear weight-sharing layers involved in this operation. First, we consider the expand setting, i.e. the output $f_{\boldsymbol{\theta}}(\boldsymbol{X}_n)$ is not reduced before the loss is applied.

**Simplified self-attention in the expand setting.** Since we want to understand if K-FAC-expand can be exact in this case, we first derive the Jacobians appearing in the derivation of K-FAC-expand in Equation (27) for all three involved layers, i.e. $\boldsymbol{J}_{\boldsymbol{s}_{Q,n,m}} f_{\boldsymbol{\theta}}(\boldsymbol{X}_n)_r$ for the layer with weights $\boldsymbol{W}^Q$, $\boldsymbol{J}_{\boldsymbol{s}_{K,n,m}} f_{\boldsymbol{\theta}}(\boldsymbol{X}_n)_r$ for the layer with weights $\boldsymbol{W}^K$, and $\boldsymbol{J}_{\boldsymbol{s}_{V,n,m}} f_{\boldsymbol{\theta}}(\boldsymbol{X}_n)_r$ for the layer with weights $\boldsymbol{W}^V$.

We can simply write the $r$-th row of the output of the layer with the weight matrix $\boldsymbol{W}^Q$ as a function of $\boldsymbol{s}_{Q,n,r}$ as

$$f_{\boldsymbol{\theta}}(\boldsymbol{X}_n)_r = \boldsymbol{s}_{Q,n,r}^\mathsf{T} \boldsymbol{S}_{K,n}^\mathsf{T} \boldsymbol{S}_{V,n}. \tag{46}$$

Therefore, we have

$$\boldsymbol{J}_{\boldsymbol{s}_{Q,n,r}} f_{\boldsymbol{\theta}}(\boldsymbol{X}_n)_r = \boldsymbol{S}_{V,n}^\mathsf{T} \boldsymbol{S}_{K,n} = \boldsymbol{b}_{Q,n,r}^\mathsf{T} \in \mathbb{R}^{C \times P_{K,\mathrm{out}}}, \tag{47}$$

with $C = P_{V,\mathrm{out}}$ and $\boldsymbol{b}_{Q,n,r,m} = \boldsymbol{0}$ for all $m \neq r$, which is the first assumption necessary for Lemma 3 to hold. While $\boldsymbol{b}_{Q,n,r}$ is not the same for all $n \in \{1, \ldots, N\}$, it is the same for all $r \in \{1, \ldots, R\}$ and hence, under the same assumptions as in Proposition 4, K-FAC-expand is exact for the layer with weights $\boldsymbol{W}^Q$ in the special case of a single data point, $N = 1$.

For the other two involved linear layers, we cannot express the $r$-th row of $f_{\boldsymbol{\theta}}(\boldsymbol{X}_n)$ as a function of the $r$-th row of $\boldsymbol{S}_{K/V,n}$, i.e. elements from all rows of $\boldsymbol{S}_{K/V,n}$ contribute to the $r$-th row of the output matrix. We can also see this by directly deriving $\boldsymbol{J}_{\boldsymbol{s}_{K/V,n,m}} f_{\boldsymbol{\theta}}(\boldsymbol{X}_n)_r$ which will be generally non-zero and dependent on $r$ and $m$. We omit the explicit derivation by taking the partial derivatives and directly state the results. For the second layer with weight matrix $\boldsymbol{W}^K$, we have

$$\boldsymbol{J}_{\boldsymbol{s}_{K,n,m}} f_{\boldsymbol{\theta}}(\boldsymbol{X}_n)_r = \boldsymbol{s}_{V,n,m} \boldsymbol{s}_{Q,n,r}^\mathsf{T} \in \mathbb{R}^{C \times P_{Q,\mathrm{out}}}. \tag{48}$$

Moreover, for the third layer with weight matrix $\boldsymbol{W}^V$, we have

$$\boldsymbol{J}_{\boldsymbol{s}_{V,n,m}} f_{\boldsymbol{\theta}}(\boldsymbol{X}_n)_r = \boldsymbol{s}_{Q,n,r}^\mathsf{T} \boldsymbol{s}_{K,n,m} \boldsymbol{I}_C \in \mathbb{R}^{C \times C}. \tag{49}$$

This means that the assumption of Lemma 3 that the $R$ elements along the weight-sharing dimension are independent does not hold, since the Jacobians depend on $r$ and $m$. The approximation leads to an inexact computation, even though only linear layers are involved and a Gaussian likelihood is used. Similarly, we can inspect the corresponding reduce case.

**Simplified self-attention in the reduce setting.** Assuming we use a scaled sum $z$ with factor $c$ as the aggregation function, we can further rewrite the Jacobians occurring in Equation (39) as

$$\begin{aligned}
\boldsymbol{J}_{\boldsymbol{\theta}_\ell} z(f_{\boldsymbol{\theta}}(\boldsymbol{X}_n)) &= \sum_{r=1}^R \boldsymbol{J}_{\boldsymbol{s}_{\ell,n,r}} z(f_{\boldsymbol{\theta}}(\boldsymbol{X}_n)) \boldsymbol{J}_{\boldsymbol{\theta}_\ell} \boldsymbol{s}_{\ell,n,r} \\
&= \sum_{r=1}^R \boldsymbol{a}_{\ell,n,r} \otimes \boldsymbol{b}_{\ell,n,r} \\
&= \sum_{r=1}^R \boldsymbol{a}_{\ell,n,r} \otimes \left( c \sum_{m=1}^R \boldsymbol{J}_{\boldsymbol{s}_{\ell,n,r}} f_{\boldsymbol{\theta}}(\boldsymbol{X}_n)_m \right),
\end{aligned} \tag{50}$$

where $\boldsymbol{J}_{\boldsymbol{s}_{\ell,n,r}} f_{\boldsymbol{\theta}}(\boldsymbol{X}_n)_m$ are the same Jacobians we have derived for the expand case. Since according to Lemma 5 we need all $\boldsymbol{b}_{\ell,n,r}$ to be the same for all $r \in \{1, \ldots, R\}$ for the first approximation to be exact under the assumptions of Proposition 6, K-FAC-reduce is only exact when $N = 1$ and only for the layer with weights $\boldsymbol{W}^Q$ – just as K-FAC-expand in the expand setting.

When we extend this scenario to a network consisting of $L$ blocks as defined in Equation (45), the above statements regarding the special case where K-FAC-expand and K-FAC-reduce are exact for the layer with weights $\boldsymbol{W}^Q$ only hold for the *last* block. While we omit an explicit derivation, intuitively, this can be seen by the fact that we cannot rewrite the $r$-th row of this model's output as a function of only the $r$-th row of the layer's output $\boldsymbol{S}_{\ell,Q,n}$ of all layers with weights $\boldsymbol{W}_\ell^Q$, besides for the layer in the last block, i.e. the layer in the $L$-th block with weights $\boldsymbol{W}_L^Q$.

This shows that even without explicit nonlinear activation functions, the self-attention mechanism in transformer models breaks the two approximations. Hence, it is not inherently clear how useful it is to consider the corresponding approximation in the expand and reduce setting. This is especially relevant given that we know that the computational complexity of K-FAC-reduce is generally smaller than of K-FAC-expand and given the similar downstream optimisation performance we report in Section 4.

### B.3.2   K-FAC for GNNs

Beyond transformers, we have introduced GNNs as a class of models that also use linear weight-sharing layers. There are many types of GNNs and we will only explicitly cover two of them here.

**Related work: node classification with GCN.** We first consider a GCN layer, described in Appendix A. This specification of K-FAC for GNNs has been previously derived for semi-supervised node classification in Izadi et al. (2020) and we include it for completeness, since it is, to the best of our knowledge, the only case where K-FAC has been applied to a GNN. The only difference to the normal derivation of K-FAC is that the inputs $\tilde{\boldsymbol{a}}_{\ell,n}$ to the $\ell$-th layer for the node with index $n$ now depend on its neighbourhood $\mathcal{N}(n)$, since

$$\tilde{\boldsymbol{a}}_{\ell,n} := \sum_{j \in \mathcal{N}(n)} \hat{\boldsymbol{C}}_{nj} \boldsymbol{a}_{\ell,j} \in \mathbb{R}^{P_{\ell,\text{in}}}. \tag{51}$$

Using this notation, the definition of the K-FAC GGN for node classification is simply

$$
\begin{aligned}
GGN(\boldsymbol{\theta}_\ell) &= \sum_{n=1}^N \boldsymbol{J}_{\boldsymbol{\theta}_\ell}(\boldsymbol{X})_n^{\mathsf{T}} \boldsymbol{\Lambda}(f_{\boldsymbol{\theta}}(\boldsymbol{X})_n) \boldsymbol{J}_{\boldsymbol{\theta}_\ell}(\boldsymbol{X})_n \\
&\approx \underbrace{\Big[\frac{1}{N} \sum_{n=1}^N \tilde{\boldsymbol{a}}_{\ell,n} \tilde{\boldsymbol{a}}_{\ell,n}^{\mathsf{T}}\Big]}_{=:\tilde{\boldsymbol{A}}_\ell} \otimes \underbrace{\Big[\sum_{n=1}^N \tilde{\boldsymbol{b}}_{\ell,n} \boldsymbol{\Lambda}(f_{\boldsymbol{\theta}}(\boldsymbol{X})_n) \tilde{\boldsymbol{b}}_{\ell,n}^{\mathsf{T}}\Big]}_{=:\tilde{\boldsymbol{B}}_\ell},
\end{aligned}
\tag{52}
$$

where $\boldsymbol{X} \in \mathbb{R}^{N \times D}$, $\tilde{\boldsymbol{b}}_{\ell,n} := \boldsymbol{J}_{\tilde{\boldsymbol{s}}_{\ell,n}} f_{\boldsymbol{\theta}}(\boldsymbol{X})_n^{\mathsf{T}}$, and $\tilde{\boldsymbol{s}}_{\ell,n} := \boldsymbol{W}_\ell \tilde{\boldsymbol{a}}_{\ell,n}$. Again, it is important to note that we need to have access to the whole neighbourhood of $\boldsymbol{x}_n$ to be able to write the $n$-th term of the GGN, which is why the input to the model is the matrix $\boldsymbol{X}$ containing all nodes for each loss term. Also, depending on the sparsity of $\hat{\boldsymbol{C}}$, i.e. the size of neighbourhoods, we might have multiple identical terms. In the extreme case of all neighbourhoods being the same, e.g. in the case of a fully connected graph, i.e. are values of $\hat{\boldsymbol{C}}$ are the same, all terms of the GGN will be the same.

According to the first of the three cases in Section 3.1, we do not need to think in terms of the expand and reduce settings here – as opposed to the case of the GraphNetwork we consider next, because we aggregate over each node's neighbourhood *before* the forward pass through a linear layer.

**Graph classification with GraphNetwork.** Now, we want to look at a more general architecture, an instance of the GraphNetwork introduced in Battaglia et al. (2018) and described in Appendix A. It is important to note that while the inputs and the GraphNetwork block structure look different from our standard input and linear layer, this case can be treated the same. This architecture is therefore a good didactic example of how to apply the presented framework of thinking about K-FAC for linear weight-sharing layers to new model architectures. In contrast to the original description in Battaglia

et al. (2018), we already defined the inputs to a GraphNetwork block according to our definition of an input that leads to weight-sharing, i.e. with an additional weight-sharing dimension of size $R$. This is in fact the crucial step to be able to apply our framework in this setting. As noted in Appendix A, the inputs cannot be trivially batched in this formulation. This is not an issue for our derivation, but it requires special consideration in the implementation, which we will consider in Appendix B.4.

First, we note that we consider the task of graph classification. Hence, our loss has the same form as Equation (36), which means that the weight-sharing dimensions have to be reduced at some point during the forward pass and we are in the reduce setting. Notably, this is the setting of the ogbg-molpcba workload of the `AlgoPerf` benchmark from the experiment in Section 4.2. Following this line of reasoning, we would simply have to apply the corresponding K-FAC approximation. Since the inputs take a more complex form than in our description of the reduce case, we still have to adopt the notation from Appendix A to concretely write down the approximation.

To recap, the weight-sharing dimension of size $R_n$ of graph $\mathbb{X}_n^G$ depends on the input graph itself (indicated by the index $n$) and which update function within a GraphNetwork block we want to derive K-FAC-reduce for. For $\phi^E$ this dimension is going to be $R_n = N_n^E$, whereas it will be $R_n = N_n^V$ for $\phi^V$. In the case of $\phi^u$ we do not have a weight-sharing dimension, as it has been reduced before this layer is applied, and we can simply apply the regular K-FAC approximation. We can define the inputs to layer $\ell$ of type $\phi^E$ as

$$\boldsymbol{A}_{\ell,n} = \operatorname{concat}(\boldsymbol{X}_n^E, \boldsymbol{X}_{n,\boldsymbol{r}_n}^V, \boldsymbol{X}_{n,\boldsymbol{s}_n}^V, \operatorname{repeat}_{N_n^E}(\boldsymbol{x}_n^u)) \in \mathbb{R}^{N_n^E \times (D_E + 2D_V + D_u)} \tag{53}$$

and as

$$\boldsymbol{A}_{\ell,n} = \operatorname{concat}(\boldsymbol{X}_n^V, \tilde{\boldsymbol{X}}_n^E, \operatorname{repeat}_{N_n^V}(\boldsymbol{x}_n^u)) \in \mathbb{R}^{N_n^V \times (D_V + D_E + D_u)} \tag{54}$$

for $\phi^V$. Correspondingly, we have $\boldsymbol{b}_{\ell,n} = \boldsymbol{J}_{\boldsymbol{S}_{\ell,n}} f_{\boldsymbol{\theta}}(\mathbb{X}_n^G)^{\mathsf{T}} \in \mathbb{R}^{N_n^E \times D_E \times C}$ for $\phi^E$ and $\boldsymbol{b}_{\ell,n} = \boldsymbol{J}_{\boldsymbol{S}_{\ell,n}} f_{\boldsymbol{\theta}}(\mathbb{X}_n^G)^{\mathsf{T}} \in \mathbb{R}^{N_n^V \times D_V \times C}$, with $\boldsymbol{S}_{\ell,n} = \boldsymbol{A}_{\ell,n} \boldsymbol{W}_\ell^{E\mathsf{T}} \in \mathbb{R}^{N_n^E \times D_E}$ and $\boldsymbol{S}_{\ell,n} = \boldsymbol{A}_{\ell,n} \boldsymbol{W}_\ell^{V\mathsf{T}} \in \mathbb{R}^{N_n^V \times D_V}$, respectively.

Using this notation, we can approximate the GGN for layer $\ell$, assuming its type is either $\phi^E$ or $\phi^V$, as

$$
\begin{aligned}
\mathbf{GGN}(\boldsymbol{\theta}_\ell) &= \sum_{n=1}^N \boldsymbol{J}_{\boldsymbol{\theta}_\ell}(\mathbb{X}_n^G)^{\mathsf{T}} \boldsymbol{\Lambda}(f_{\boldsymbol{\theta}}(\mathbb{X}_n^G)) \boldsymbol{J}_{\boldsymbol{\theta}_\ell}(\mathbb{X}_n^G) \\
&\approx \sum_{n=1}^N \underbrace{\left[\sum_{r=1}^{R_n} \frac{1}{\sqrt{R_n}} \boldsymbol{a}_{\ell,n,r}\right]}_{=:\hat{\boldsymbol{a}}_{\ell,n}} \otimes \underbrace{\left[\sum_{r=1}^{R_n} \frac{1}{\sqrt{R_n}} \boldsymbol{b}_{\ell,n,r}\right]}_{=:\hat{\boldsymbol{b}}_{\ell,n}} \boldsymbol{\Lambda}(f_{\boldsymbol{\theta}}(\mathbb{X}_n^G)) \left[\sum_{r=1}^{R_n} \frac{1}{\sqrt{R_n}} \boldsymbol{a}_{\ell,n,r}\right] \otimes \left[\sum_{r=1}^{R_n} \frac{1}{\sqrt{R_n}} \boldsymbol{b}_{\ell,n,r}\right]^{\mathsf{T}} \\
&\approx \underbrace{\left[\frac{1}{N} \sum_{n=1}^N \hat{\boldsymbol{a}}_{\ell,n} \hat{\boldsymbol{a}}_{\ell,n}^{\mathsf{T}}\right]}_{=:\hat{\boldsymbol{A}}_\ell} \otimes \underbrace{\left[\sum_{n=1}^N \hat{\boldsymbol{b}}_{\ell,n} \boldsymbol{\Lambda}(f_{\boldsymbol{\theta}}(\mathbb{X}_n^G)) \hat{\boldsymbol{b}}_{\ell,n}^{\mathsf{T}}\right]}_{=:\hat{\boldsymbol{B}}_\ell},
\end{aligned}
\tag{55}
$$

analogously to Equation (39). However, there is one important difference: since $R_n$ now depends on the $n$-th data point, it makes a difference where we insert the scaling by $1/R_n$, since $\hat{\boldsymbol{A}}_\ell$ and/or $\hat{\boldsymbol{B}}_\ell$ are now weighted sums. To avoid having one potentially non-uniformly weighted and one unweighted sum, we choose to not just include the scaling by $1/R_n$ in $\hat{\boldsymbol{a}}_{\ell,n}$ as in Equation (39), or in $\hat{\boldsymbol{b}}_{\ell,n}$. Instead, to try to keep the overall scale and the weighting of the $N$ terms balanced, we simply include a scaling by $1/\sqrt{R_n}$ in *both*, $\hat{\boldsymbol{a}}_{\ell,n}$ and $\hat{\boldsymbol{b}}_{\ell,n}$.

## B.4 Practical Considerations

While we have discussed theoretically how to apply K-FAC to linear weight-sharing layers, we now turn to implementation details and computational considerations that are crucial for the practical application of the approximations.

**Implementation details.** There are multiple libraries that implement K-FAC for popular deep learning frameworks like `Jax` (Bradbury et al., 2018) and `PyTorch` (Paszke et al., 2019), e.g. `KFAC-JAX`

```
1   # Check if there even is a weight-sharing dimension; if not, the Kronecker
2   # factors can directly be calculated.
3   if in_data.ndim == 3:
4       # Mini-batch size M, weight-sharing dimension R, feature dimension P_{ℓ,in/out}.
5       M, R, P_in = in_data.shape
6       P_out = out_grads.shape[2]
7       if approximation == 'expand':
8           # Flatten the weight-sharing dimension into the mini-batch dimension.
9           in_data = in_data.view(M*R, P_in) / math.sqrt(R)
10          out_grads = out_grads.view(M*R, P_out)
11      elif approximation == 'reduce':
12          # Reduce the weight-sharing dimension with mean and sum.
13          in_data = in_data.mean(dim=1)
14          out_grads = out_grads.sum(dim=1)
15  # Calculate Kronecker factors A_ℓ/Â_ℓ and B_ℓ/B̂_ℓ.
16  A = torch.matmul(in_data.T, in_data) / M
17  B = torch.matmul(out_grads.T, out_grads)
```

Listing 1: **Illustration of K-FAC-expand and K-FAC-reduce with code.** This piece of code calculates the approximations on one mini-batch and for one layer. We receive the inputs to the layer, `in_data`, from a forward hook and the gradients of the loss w.r.t. the outputs of the layer, `out_grads`, from a backward hook. Here we assume that only one additional weight-sharing dimension exists and that the first dimension is always the mini-batch dimension. Note that the dimensions of `in_data` and `out_grads` are the same, *independently of the setting we are in* (c.f. Section 3.1). This illustrates why we can choose to use either approximation, K-FAC-expand or K-FAC-reduce, in each setting. The actual implementation in ASDL is very similar for `torch.nn.Linear` modules, the logic is just separated into multiple functions and allows for multiple weight-sharing dimensions. However, the adjustments for the GraphNetwork are more involved.

(Botev & Martens, 2022) for `Jax` and `BackPACK` (Dangel et al., 2020), `ASDL` (Osawa et al., 2023), and `KFAC-PyTorch` (Pauloski et al., 2021) for `PyTorch`. We focus on the implementation of K-FAC-expand and K-FAC-reduce within `ASDL`, which we also use for the experiments in Section 4. K-FAC is implemented using forward and backward hooks, which allow us to get the inputs to a specific layer and the gradients of the loss w.r.t. the layer outputs – which are the ingredients we need for all K-FAC approximations. Notably, this requires that linear layers are implemented with `torch.nn.Linear` instances, since otherwise, the implementation with hooks does not work. The default implementation of multi-head attention in `PyTorch` does indeed not use the required linear modules, so the implementation has to be adjusted to work with common K-FAC implementations like `ASDL`. In contrast, other methods like Shampoo (Gupta et al., 2018) and Tensor Normal Training (Ren & Goldfarb, 2021) are agnostic to the architecture. Assuming the implementation of the model is appropriate, K-FAC-expand and K-FAC-reduce in their simplest form only require a minor adjustment in the code base for regular K-FAC, which is presented in Listing 1.

However, if we wanted to use K-FAC-expand in the expand and K-FAC-reduce in the reduce setting, we would need to find a way of automatically determining the setting we are in. For all models considered here, i.e. the (vision) transformer and GNN, only one of the two settings applies to all linear layers with weight-sharing. Hence, using a single additional forward pass, we could check if any linear weight-sharing layers are used and what the shape of the model output is. From this, we can deduce if the expand or the reduce case applies. As we mentioned before, this might not even be desirable, as it is unclear if we should always use the approximation theoretically motivated by the setting. Alternatively, a single flag set by the user can determine if K-FAC-expand or K-FAC-reduce is applied to all linear weight-sharing layers.

This implementation obviously assumes that we even have an explicit weight-sharing dimension. In the case of K-FAC-reduce for the GraphNetwork introduced in Appendix A, we have to adopt our implementation due to the batching technique that is employed for graph inputs in practice. Since each graph in a mini-batch $\mathcal{M}$ of size $M$ might have a different weight-sharing dimension $R_m$, i.e. the number of nodes and the number of edges of each graph, we cannot batch them trivially. As a solution, the inputs for each graph as stated in Equation (53) and Equation (54) are simply concatenated in the first dimension, which results in a dimension of size $R_{\mathcal{M}} := \sum_{m=1}^{M} R_m$. To apply K-FAC-expand here, we do not have to modify anything, besides scaling the approximation for

each mini-batch by $1/R_{\mathcal{M}}$ instead of $1/M$.[8] To apply K-FAC-reduce, we can use a *scatter mean*, which aggregates tensor elements according to indices, to implement the mean (with the square root scaling from Equation (55)) operation without having an explicit weight-sharing dimension. Unfortunately, this creates two issues. First, we have to know that this adjustment to K-FAC is even required for a specific layer since we cannot deduce it from the shape of the layer inputs. Second, the scatter mean requires additional information, since we need to know to which graphs the nodes/edges in the input belong. One approach to resolve these issues is to define a custom layer type for this type of linear layer, which has an attribute containing the indices of all nodes/edges for each graph in the batch. However, this requires changes to the model architecture implementation, because additional attributes have to be set for the regular linear modules.

Besides the changes necessary for K-FAC-expand and K-FAC-reduce, we can use the same additional algorithmic tools often used for optimisation with K-FAC. Typically, *damping* is used (Martens & Grosse, 2015), i.e. a scalar is added to the diagonal of the two Kronecker factors $\boldsymbol{A}$ and $\boldsymbol{B}$ or the diagonal of their product – the latter corresponds to adding the Hessian of an isotropic Gaussian prior over the weights. Also, since we usually operate in the stochastic setting and only compute the K-FAC approximation on mini-batches, sometimes an exponential moving average over the Kronecker factors is used. However, in our experiments, we do not use such a moving average and only compute K-FAC for a single mini-batch (besides for Figure 5, as mentioned in Appendix C.4) and still reduce the update frequency of the K-FAC statistics and the preconditioner.

**Computational considerations.** Besides the implementation details, we also have to consider the computational cost when deploying K-FAC approximations in practice. Here, we have to respect the same constraints as with regular K-FAC. When we have a large output dimension $C$, it is expensive or even unfeasible to propagate the $C \times C$ loss Hessian $\boldsymbol{H}_{f_{\boldsymbol{\theta}}}\ell(\boldsymbol{y}_n, f_{\boldsymbol{\theta}}(\boldsymbol{x}_n))$ for each of the $N$ data points through the computation graph. Instead, we use the fact that we have

$$\mathbb{E}_{\boldsymbol{y}\sim p(\boldsymbol{y}|f_{\boldsymbol{\theta}}(\boldsymbol{x}_n))}[\nabla_{f_{\boldsymbol{\theta}}} \log p(\boldsymbol{y}|f_{\boldsymbol{\theta}}(\boldsymbol{x}_n))\nabla_{f_{\boldsymbol{\theta}}} \log p(\boldsymbol{y}|f_{\boldsymbol{\theta}}(\boldsymbol{x}_n))^{\mathsf{T}}] = \boldsymbol{H}_{f_{\boldsymbol{\theta}}}\ell(\boldsymbol{y}_n, f_{\boldsymbol{\theta}}(\boldsymbol{x}_n)) \qquad (56)$$

and take $S$ Monte Carlo (MC) samples from the model's predictive distribution $\boldsymbol{y}_s \sim p(\boldsymbol{y}|f_{\boldsymbol{\theta}}(\boldsymbol{x}_n))$. Taking a single sample results in a rank-1 MC approximation of the true loss Hessian and only requires the propagation of a single vector through the computation graph for each data point.

## C  Additional experimental details and results

### C.1  MLCommons' `AlgoPerf` benchmark for training algorithms

The goal of the `AlgoPerf` benchmark for training algorithms (Dahl et al., 2023) is to measure "training speedups due to algorithmic improvements" (MLCommons, 2022). Specifically, the benchmark defines multiple workloads, where one workload is defined by the combination of a dataset, a neural network model, a loss function, and a target performance defined by some evaluation metric. For example, training a ViT on ImageNet using the cross-entropy loss until a target validation accuracy of 77.309% has been reached constitutes a workload. In the `AlgoPerf` benchmark, training algorithms are compared on fixed hardware in terms of the wall-clock time they require to reach the fixed validation target performance.

For our experiments, we use two of the workloads of the `AlgoPerf` benchmark as realistic deep learning problems, to showcase the optimisation behaviour of our two K-FAC variations (Sections 4.2 and 4.3). Similar to the benchmark, we measure both the number of steps and the wall-clock time necessary to reach the target validation metric. To put the training behaviour of K-FAC-expand and K-FAC-reduce into perspective, we compare them to the *target-setting* run of the `AlgoPerf` benchmark, denoted *reference run* in the experiments (Sections 4.2 and 4.3). To determine the fixed validation target for each workload, four standard algorithms (AdamW, NAdamW, SGD with Nesterov momentum (Nesterov), and SGD with heavy ball momentum) were each tuned with a budget of 200 trials for every workload. The combination of algorithm and hyperparameters that reached the highest validation performance within a fixed time was then run 20 times and the median of the best achieved validation metric was set as the validation target. As our reference run, we use the workload-specific, best-performing algorithm and hyperparameter combination. Note that this should

---

[8]As explained below Equation (55), we could also choose a different way of scaling here. We choose this one as it enables a simple implementation in contrast to K-FAC-reduce.

Table 3: Results from Table 1 with standard errors over 10 epochs.

| K-FAC | Batch size | | | | |
|---|---|---|---|---|---|
| | 128 | 256 | 512 | 1024 | 2048 |
| expand | $0.24 \pm 0.01$ | $0.38 \pm 0.02$ | $0.75 \pm 0.03$ | $1.36 \pm 0.05$ | OOM |
| reduce | $0.17 \pm 0.01$ | $0.24 \pm 0.02$ | $0.43 \pm 0.03$ | $0.63 \pm 0.04$ | $1.17 \pm 0.08$ |

not be confused with the baseline algorithms in the `AlgoPerf` benchmark, since the target-setting runs are tuned for the best possible validation performance on each workload.

K-FAC is run with most of the hyperparameters of the reference run and a tuning budget of 15 runs for tuning the learning rate and damping. This means that most hyperparameters are not directly tuned for K-FAC. To enable a fairer comparison, we also use the same additional budget for tuning the learning rate of the reference run, the same learning rate schedule, and tuning goal as K-FAC. However, none of these runs reaches the validation metric target, which is why the reference runs correspond to the original target-setting runs. Despite this, the experiments presented in Sections 4.2 and 4.3 are *not* meant to demonstrate that K-FAC is a superior training algorithm. To provide actual evidence for this claim we would have to run a *valid* and optimised submission on the entire `AlgoPerf` benchmark or a benchmark of comparable rigour. Nevertheless, using the well-tuned reference run allows us to put K-FAC's training performance into perspective.

The two `AlgoPerf` workloads used in this paper are:

**Graph neural network on ogbg-molpcba.** The workload consists of a (binary) cross-entropy loss, the GraphNetwork instance described in Appendix A, and the ogbg-molpcba molecular property prediction dataset (Hu et al., 2020). Each molecule is represented as a graph, where atoms are nodes and edges are chemical bonds. The task is to predict whether or not a molecule has certain chemical properties; there are 128 different properties. For training, we have about 350k examples and almost 44k for validation and testing. The validation target mean average precision (mAP) determined by the target-setting runs is 0.28098 and the best performing reference run algorithm is Nesterov.

**Vision transformer on ImageNet.** This workload also uses the cross-entropy loss, a vision transformer architecture, and the LSVRC-2012 ImageNet (short: ImageNet) image dataset (Russakovsky et al., 2015). The goal is to classify images into one of 1000 classes. There are about 1.3 million training, 50k validation, and 10k test examples. The validation target accuracy determined by the target-setting runs is 0.77309 and the best performing reference run algorithm is NAdamW.

### C.2 Update step speed with K-FAC-expand and K-FAC-reduce

We provide the full results of the update step timing experiment with standard errors over 10 epochs in Table 3. The model architecture, optimiser, and data setup are exactly the same as described in Appendix C.5. The full results of the GPT-2 (`nanoGPT`) timing experiment are presented in Table 4; the mean and standard error are computed based on three runs.

Table 4: Timing of K-FAC GGN approximation for GPT-2 on full DART dataset.

| K-FAC | Absolute time [min] ↓ | Relative time [%] ↓ |
|---|---|---|
| expand | $9.55 \pm 0.13$ | 100 |
| reduce | $6.68 \pm 0.04$ | 70 |

### C.3 Graph neural network on ogbg-molpcba

For this workload, we use a training batch size of 512 and a single NVIDIA V100 32GB GPU for each run. The reference run algorithm (Nesterov) uses a learning rate of 2.4917728606918423, $\beta_1$ equal to 0.9449369031171744, weight decay set to $1.2859640541025928e{-7}$, and linear warmup for 3,000 steps and a polynomial schedule with a decay steps factor of 0.861509027839639 and an end factor of $1e{-3}$. The two K-FAC variations use the exact same hyperparameters, but the warmup and expected number of steps (60,000) are multiplied by 0.75 and the learning rate and the damping are tuned via random search. The search space for the learning rate is log uniform values in $[0.1, 10]$ and for the damping in $[1e{-3}, 1]$. We choose the hyperparameters of the run that first reaches the validation target. This setting is a learning rate of 9.96871902194967 and damping of 0.7881965339190345 for K-FAC-expand and a learning rate of 0.5885756514016359 and damping

of 0.0579230193904011 for K-FAC-reduce. We then repeated each run five times. The Kronecker factors and the preconditioner are computed every 10 iterations and we use a single sample MC approximation of the Fisher, as explained in Appendix B.4.

### C.4 Vision transformer on ImageNet

We use a training batch size of 1,024, $\epsilon = 1e-8$ for NAdamW, and $4\times$ NVIDIA V100 32 GPUs for all runs on this workload. The reference run algorithm (NAdamW) is using a learning rate of 0.0008445074561975979, $\beta_1$ equal to 0.8895758153482813, $\beta_2$ to 0.9978504782314613, weight decay set to 0.08135402759553023, linear warmup for 6999 steps, and a cosine decay schedule. The two K-FAC variations use the exact same hyperparameters, but the warmup and expected number of steps (140,000) is multiplied by 0.75 and the learning rate and the damping are tuned via random search. The search space for the learning rate and the damping is log uniform values in $[1e-4, 1e-2]$. We choose the hyperparameters of the run that first reaches the validation target. For both K-FAC variations, the best setting is a learning rate of 0.0012662938340704357 and damping of 0.00016524019235426572. The Kronecker factors and the preconditioner are computed every 50 iterations and we use a single sample MC approximation of the Fisher, as explained in Appendix B.4.

We also conduct a run where we update the Kronecker factors every step, use an exponential moving average with a factor (`ema_decay` in ASDL) equal to $\beta_2$, update the preconditioner every 10 steps, set the damping to $1e-5$, and use all the other hyperparameters of a reference run setting, including the learning rate. The reference run setting corresponds to an earlier target-setting run result from the `AlgoPerf` repository and uses a learning rate of about $2e-3$, $\beta_1 = 0.7132$, $\beta_2 = 0.9982$, and the weight decay is set to 0.026595. Moreover, it clips the gradients to keep their norm below 1. As before, we also multiply the number of warmup and expected number of steps by 0.75. Due to the high update frequency of the Kronecker factors, we can see the significant wall-clock time difference between K-FAC-expand and K-FAC-reduce in Figure 5. Both variations are similar in terms of steps to the target, K-FAC-expand takes about 92.6k and K-FAC-reduce takes about 93.7k steps, but whereas K-FAC-expand takes about 50 hours, K-FAC-reduce reaches the target after about 37 hours. Note, that both variations are still significantly slower than the NAdamW reference run which only takes about 25 hours, but 117.4k steps.

### C.5 K-FAC for automatic hyperparameter selection via marginal likelihood optimisation

K-FAC is used for the Hessian approximation within the Laplace approximation by first accumulating it over the entire data set and using the eigendecomposition of Kronecker factors as in Immer et al. (2021). This is done every five epochs during training to update the weight decay parameters per layer of the neural network. Therefore, the computation time of K-FAC makes a significant contribution to the overall training time: reduce takes on average 75% of the overall time of expand, a result of reducing the wall-clock time of a single marginal likelihood update step by about 50%, as shown in Table 2. Here, we provide additional experimental details and the full table with standard errors.

For the marginal likelihood optimisation experiment, we use the same settings as Daxberger et al. (2021), i.e., a Wide ResNet 16-4 with Fixup initialisation and parameters instead of batch normalisation (Zagoruyko & Komodakis, 2016; Zhang et al., 2019b). We use a batch size of 128, an initial learning rate of 0.1 cosine decayed to $1e-6$ and weight decay of $5e-3$ training for 100 epochs. We use standard data augmentation with horizontal flips and random crops. Results are averaged over three seeds and the timings are obtained running all runs on a single NVIDIA A100 GPU and locally measuring the hyperparameter update time. The full results with standard errors are presented in Table 5.

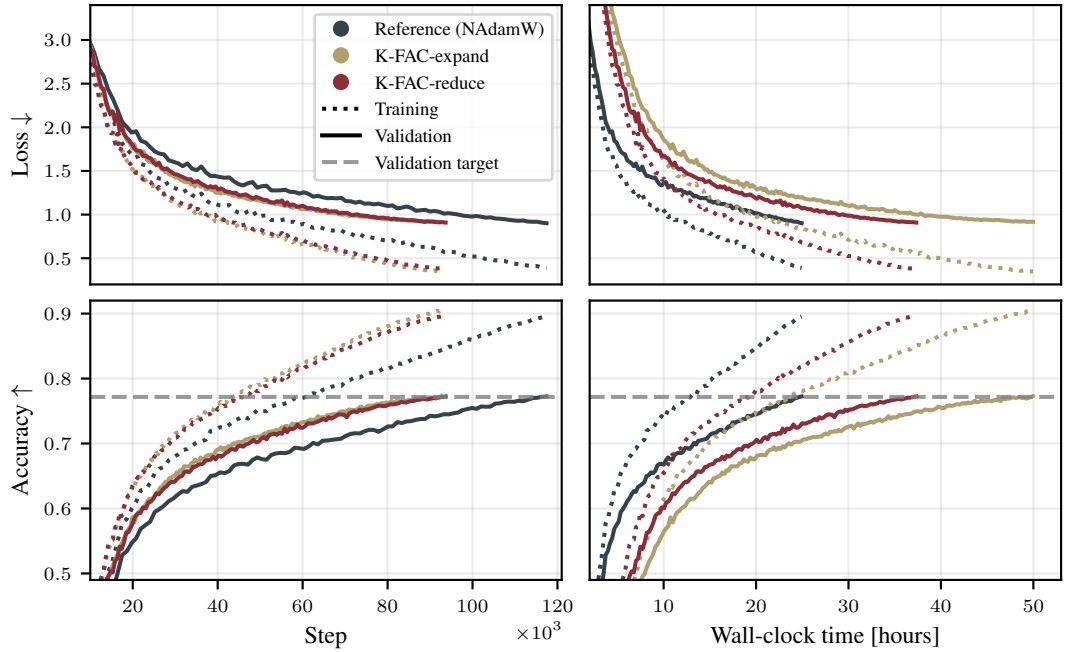

Figure 5: **Training results for a vision transformer on ImageNet.** This is similar to Figure 4, but the K-FAC statistics are updated *every* step and different hyperparameters are used. Due to K-FAC's overhead, the wall-clock time is not reduced in this setting. Moreover, the discrepancy in speed between K-FAC-expand and K-FAC-reduce becomes apparent.

Table 5: Results from Table 2 with standard errors over three random seeds. The update time is the average time per full K-FAC Laplace approximation to the marginal likelihood, which requires a full data set pass with computation and eigendecomposition of the Kronecker factors.

| K-FAC | Data Augmentation | NLL $\downarrow$ | Accuracy [%] $\uparrow$ | Update Time [s] $\downarrow$ |
|---|---|---|---|---|
| expand | ✗ | $0.422 \pm 0.013$ | $88.89 \pm 0.24$ | $196 \pm 22$ |
|  | ✓ | $0.244 \pm 0.004$ | $92.52 \pm 0.12$ |  |
| reduce | ✗ | $0.703 \pm 0.012$ | $86.71 \pm 0.13$ | $99 \pm 23$ |
|  | ✓ | $0.352 \pm 0.008$ | $93.50 \pm 0.06$ |  |

