# OpenReview forum: "Kronecker-Factored Approximate Curvature for Modern Neural Network Architectures"
_NeurIPS.cc/2023/Conference — NeurIPS 2023 spotlight_

### Official Review · Reviewer_hvpV · 2023-07-05

**Soundness:** 3 good
**Presentation:** 3 good
**Contribution:** 3 good
**Rating:** 7
**Confidence:** 3

**Summary:**

This paper extends Kronecker-Factored Approximate Curvature (K-FAC) into training generic neural networks, especially for transformers and graph neural networks. This is done by taking advantage of the weight sharing mechanism. The authors propose two flavors of K-FAC (expand and reduce), depending on the dimension of the output tensor. The authors prove that they are exact for deep linear networks. The proposed methods reach a validation metric target of a well-tuned first-order baseline with smaller steps and less wall-clock time.

**Strengths:**

1. This paper extends K-FAC into transformers and graph neural networks. Although some previous works try to extend K-FAC into transformers, their approximation is inexact, as pointed by the authors.

2. The proposed methods have mathematical foundations. Because the authors prove the exactness under deep linear settings.

3. The proposed methods reach a validation metric target of a well-tuned first-order baseline with smaller steps and less wall-clock time. So it is possible to replace first order methods.

4. This paper provides sufficient supplementary material, including mathematical derivations, additional experiments, and reproduction codes.


**Weaknesses:**

1. The performance of transformers is evaluated on ImageNet. Since transformers are mostly used for language modeling, it is better to evaluate your optimizers on some language tasks (a very big dataset is unnecessary).

2. The authors prove the exactness under deep linear settings. If possible, It is better to provide some error bounds for non-linear settings.

3. It is helpful to provide the time complexity (not wall-clock time) and space complexity for both K-FAC-expand and K-FAC-reduce.

**Questions:**

In figure 4, the K-FAC statistics are updated every 50 steps.

1. Is the same setting used in other figures?
2. What if we change the number of steps?


**Limitations:**

This paper has no negative societal impact.

---

> ### Author Rebuttal · Authors · 2023-08-09
>
> We thank the reviewer for their time and their comments.
>
> >The performance of transformers is evaluated on ImageNet. Since transformers are mostly used for language modeling, it is better to evaluate your optimizers on some language tasks (a very big dataset is unnecessary).
>
> We believe that transformers are interesting in many settings and vision transformers in particular are certainly a widely used model class -- according to the paperswithcode website on vision transformers they are currently even more popular than ResNets and the original paper proposing them has over 19k citations. Moreover, K-FAC(-expand) has been applied to transformers on language tasks before [1][2][3], but never to a vision transformer.
> However, we fully agree that it would be great to have results for a language task. To demonstrate the applicability of K-FAC to language models, **we compute a K-FAC GGN approximation with nanoGPT, a popular implementation of GPT-2, on the full DART dataset** [4]. In the following table we present the timing results of the whole computation for K-FAC-expand (currently used for transformers) and K-FAC-reduce. In line with the complexities and our other results K-FAC-reduce is significantly faster, only taking 70% of the time of K-FAC-expand. We show the mean and standard error of the timing of three runs.
> | K-FAC  | Absolute time (minutes) | Relative time (%) |
> |--------|-------------------------|-------------------|
> | expand | 9.55 +/- 0.13           | 100               |
> | reduce | 6.68 +/- 0.04           | 70                |
>
>
> >The authors prove the exactness under deep linear settings. If possible, It is better to provide some error bounds for non-linear settings.
>
> We use the exactness in the deep linear setting as motivation for our derivation, analogous to the original derivation of K-FAC for regular linear layers [5][6].
> We believe it is hard to derive useful error bounds for the deep non-linear setting, without making strong assumptions. Moreover, it is unclear if statements about the approximation quality translate in downstream optimisation performance in practice.
>
> >It is helpful to provide the time complexity (not wall-clock time) and space complexity for both K-FAC-expand and K-FAC-reduce.
>
> We state the time complexity of both variations in lines 242-244 in the main text of the paper. We will add the space complexity, which is $\mathcal{O}(NR(P_{\ell, \mathrm{in}} + P_{\ell, \mathrm{out}}))$ for K-FAC-expand and $\mathcal{O}(N(P_{\ell, \mathrm{in}} + P_{\ell, \mathrm{out}}))$ for K-FAC-reduce, where $N$ is the dataset size, $R$ is the size of the weight-sharing dimension (e.g. the number of tokens for language data), and $P_{\ell, \mathrm{in}}$ is the input and $P_{\ell, \mathrm{out}}$ the output dimension of the layer.
>
> >In figure 4, the K-FAC statistics are updated every 50 steps.
> >1. Is the same setting used in other figures?
> >2. What if we change the number of steps?
>
> 1. This setting is only used for Figure 4. In Figure 3 the K-FAC statistics are updated every ten steps, as described in section 4.2 of the paper. We will add this information to the caption of the figure.
> 2. In Figure 5 in the appendix you can see a run where the statistics are updated every step. There is little to no difference in the optimisation performance. For practical applications it is an interesting question how sparsely K-FAC can be updated. Also, note that sparse updates are already commonly used for second-order methods, see e.g. [7][8][9].
>
> We hope that we have sufficiently addressed your concerns and questions. Please let us know if there still is anything unclear.
>
> [1] G. Zhang, L. Li, Z. Nado, J. Martens, S. Sachdeva, G. E. Dahl, C. J. Shallue, R. Grosse. Which algorithmic choices matter at which batch sizes? Insights from a noisy quadratic model. NeurIPS, 2019.
>
> [2] J. G. Pauloski, Q. Huang, L. Huang, S. Venkataraman, K. Chard, I. T. Foster, Z. Zhang. KAISA: an adaptive second-order optimizer framework for deep neural networks. International Conference for High Performance Computing, Networking, Storage and Analysis (SC21), 2021.
>
> [3] K. Osawa, S. Li, T. Hoefler. PipeFisher: Efficient training of large language models using pipelining and Fisher information matrices. Preprint (arXiv: 2211.14133), 2022.
>
> [4] L. Nan et al. DART: Open-Domain Structured Data Record to Text Generation. NAACL-HLT, 2021.
>
> [5] J. Martens, R. Grosse. Optimizing neural networks with Kronecker-factored approximate curvature. ICML, 2015.
>
> [6] A. Bernacchia, M. Lengyel, G. Hennequin. Exact natural gradient in deep linear networks and its application to the nonlinear case. NeurIPS, 2018.
>
> [7] V. Gupta, T. Koren, Y. Singer. Shampoo: Preconditioned stochastic tensor optimization. ICML, 2018.
>
> [8] Y. Ren, D. Goldfarb. Tensor normal training for deep learning models. NeurIPS, 2021.
>
> [9] K. Osawa, S. Ishikawa, R. Yokota, S. Li, T. Hoefler. ASDL: A unified interface for gradient preconditioning in PyTorch. Preprint (arXiv: 2305.04684), 2023.

---

> > ### Comment · Reviewer_hvpV · 2023-08-18
> >
> > Thanks for your response. All my questions have been addressed, and I keep my rating.

---

### Official Review · Reviewer_NJ6c · 2023-07-07

**Soundness:** 3 good
**Presentation:** 4 excellent
**Contribution:** 3 good
**Rating:** 5
**Confidence:** 3

**Summary:**

This paper introduces two different settings of linear weight-sharing layers to fit for the second-order optimization method Kronecker-Factored Approximate Curvature (K-FAC). Experiments on GNNs and ViTs achieve less number of steps compared with the first-order baseline.

**Strengths:**

Experiments include GNNs and ViTs, also the results are quite inspiring in the acceleration of steps.

**Weaknesses:**

Fig. 4 doesn’t show the loss and accuracy variations after convergence.



**Questions:**

How about the comparison with other efficient optimization methods other than weight sharing?

**Limitations:**

The authors discussed the runtime of the two proposed schemes.

---

> ### Author Rebuttal · Authors · 2023-08-10
>
> >How about the comparison with other efficient optimization methods other than weight sharing?
>
> We believe there is a fundamental misunderstanding. **Linear weight-sharing is a property of the layers of the model architecture and not of the optimisation method.** In our paper, we show how we can apply K-FAC to model architectures with linear weight-sharing layers by leveraging their structure.
>
> >Fig. 4 doesn’t show the loss and accuracy variations after convergence.
>
> This is due to our experimental setting: as described in the experiments section, we follow the setup of the AlgoPerf benchmark and only train until a validation target performance is reached. While we think that it is interesting to investigate the behaviour of K-FAC at convergence, this is an orthogonal question and requires a different experimental setup. Since we did not tune for the best validation performance at convergence, running the experiments for more steps will not allow us to make any statement about the performance at convergence. E.g., the learning rate schedule will stop adopting the learning rate soon after the target is reached.

---

### Official Review · Reviewer_T2nD · 2023-07-08

**Soundness:** 3 good
**Presentation:** 2 fair
**Contribution:** 2 fair
**Rating:** 5
**Confidence:** 3

**Summary:**

This paper propose two methods called KFAC-reduce and KFAC-expand to conduct the secon-order based training for neural networks. The key idea is to approximate the Gauss–Newton matrix with low-rank matrix decomposition. The experiments on several tasks show that the propsed method can achieves faster convergence rate compared to NAdamW optimizer.

**Strengths:**

- The proposed method has well-grounded theoretical support.
- The proposed method has shown better results than the NAdamW/SGD optimizer.

**Weaknesses:**

My biggest concern comes from the evaluation sections.
- The proposed method is only verified on limited number of tasks and networks (4 configurations). The author should show more results with various configurations.
- The method is only compared with two type of optimizers. For example, SGD is the baseline for GNN experiments while NAdamW is the baseline for the ViT experiment.

**Questions:**

- Fig 4., the loss is not flat and still tend to decrease. Could the author extend the training steps until the loss curve becomes flat?

**Limitations:**

Please check the weakness sections.

---

> ### Author Rebuttal · Authors · 2023-08-10
>
> We thank the reviewer for their time and comments.
> >The proposed method is only verified on limited number of tasks and networks (4 configurations). The author should show more results with various configurations.
> >The method is only compared with two type of optimizers. For example, SGD is the baseline for GNN experiments while NAdamW is the baseline for the ViT experiment.
>
> We believe that these points are the result of a misunderstanding. The goal of our experimental evaluation is not to make any general statement about the algorithmic superiority of the K-FAC variations as an optimiser.
>
> In contrast, the goal of the paper is to generalise K-FAC to modern neural network architectures and **the proposed variations should really be understood as a generalisation of K-FAC and not an alternative method.** We use our experiments as a “proof of concept”, to show that the K-FAC variations have the potential to speed up training, to empirically validate the difference in the computational complexity of the two variations, and present an example of another use case, online approximate marginal likelihood optimisation for automatic weight decay tuning. But we fully agree that our results are very promising and hope that our work can pave the way for a rigorous comparison, e.g. in form of a submission to the AlgoPerf benchmark [1].
>
>
> >Fig 4., the loss is not flat and still tend to decrease. Could the author extend the training steps until the loss curve becomes flat?
>
> This is due to our experimental setting: as described in the experiments section, we follow the setup of the AlgoPerf benchmark and only train until a validation target performance is reached. While we think that it is interesting to investigate the behaviour of K-FAC at convergence, this is an orthogonal question and requires a different experimental setup. Since we did not tune for the best validation performance at convergence, running the experiments for more steps will not allow us to make any statement about the performance at convergence. E.g., the learning rate schedule will stop adopting the learning rate soon after the target is reached.
>
> [1] G. E. Dahl, F. Schneider, Z. Nado, N. Agarwal et al. Benchmarking Neural Network Training Algorithms. Preprint (arXiv: 2306.07179), 2023.

---

> > ### Comment · Reviewer_T2nD · 2023-08-19
> >
> > Most of my concerns are solved and I increase my rating from 4 to 5.

---

### Official Review · Reviewer_M1Nr · 2023-07-13

**Soundness:** 3 good
**Presentation:** 2 fair
**Contribution:** 3 good
**Rating:** 7
**Confidence:** 3

**Summary:**

The researchers propose a framework for applying the Kronecker-Factored Approximate Curvature (K-FAC) method to neural networks with linear weight-sharing layers. They identify two different settings of linear weight-sharing layers, leading to two variations of K-FAC: expand and reduce. They demonstrate that both variations are exact for deep linear networks with weight-sharing in their respective settings. They find that K-FAC-reduce is generally faster than K-FAC-expand and leverage this to accelerate automatic hyperparameter selection for a Wide ResNet. They also show that both K-FAC variations can achieve a target validation metric in a significantly reduced number of steps compared to a first-order baseline, resulting in a comparable improvement in training time.

**Strengths:**

Very interesting perspective by looking at modern NN building blocks as weight shared linear layers.
Solid results and thorough experiments. It's great that the authors measure actual wall-clock time speedup instead of simply recording the steps. This shows that it's a method with practical merits.

**Weaknesses:**

Minor issues in presentations, such as the plots are not really hard to read, and organization of the propositions are too clustered to be read comfortably.

**Questions:**

1. what does Exactness of K-FAC-reduce and K-FAC-expand? why is this property important?
2. How does it fare in lower precision setting, is this still numerically stable?

**Limitations:**

providing more intuition in the theory section will be highly appreciated. From the reader points of view, who are not familiar with the notation setup, it looks like a "wall of math", everyone complains about.

---

> ### Author Rebuttal · Authors · 2023-08-09
>
> We thank the reviewer for their time and appreciate that they also found the concept of linear weight-sharing layers to describe modern neural network layers useful.
>
> >Minor issues in presentations, such as the plots are (...) really hard to read, and organization of the propositions are too clustered to be read comfortably.
>
> We will improve the readability of the plots by summarising the lines for runs with different seeds and changing the colour scheme. Moreover, we will use the additional space to improve the readability of the propositions. In case you have any other concrete suggestions, we would highly appreciate it.
>
> >what does Exactness of K-FAC-reduce and K-FAC-expand? why is this property important?
>
> We use the exactness of the two K-FAC variations in their respective settings purely as motivation for their derivation, since this is exactly the original motivation used for deriving K-FAC for regular linear layers [1][2]. **The proposed variations should really be understood as a generalisation of K-FAC and not an alternative method.** The statements do not say anything about the downstream optimisation performance or even just the approximation quality in deep nonlinear networks.
>
> >How does it fare in lower precision setting, is this still numerically stable?
>
> We have not tested this experimentally, but we expect that K-FAC might encounter numerical issues to a similar extent as all other optimisation methods that have to invert dense matrices. However, there is an inverse-free natural gradient method that also relies on K-FAC, which we expect to do much better in lower precision settings [3]. Crucially, since a K-FAC approximation to the Fisher/GGN is also needed for this method, our work by extension enables this method for the here covered modern neural network architectures.
>
> >providing more intuition in the theory section will be highly appreciated.
>
> We are happy to use the additional page for more details and intuitions on the derivations. Specifically, we will add an intuitive visualisation of the the different settings K-FAC can be applied in and how this relates to the different K-FAC variations.
>
> [1] J. Martens, R. Grosse. Optimizing neural networks with Kronecker-factored approximate curvature. ICML, 2015.
>
> [2] A. Bernacchia, M. Lengyel, G. Hennequin. Exact natural gradient in deep linear networks and its application to the nonlinear case. NeurIPS, 2018.
>
> [3] W. Lin, V. Duruisseaux, M. Leok, F. Nielsen, M. E. Khan, M. Schmidt. Simplifying Momentum-based Positive-definite Submanifold Optimization with Applications to Deep Learning. ICML, 2023.

---

> > ### Comment · Reviewer_M1Nr · 2023-08-16
> >
> > Thank you for responding to my questions. Paper with good insights! I am keeping my rating as accept.

---

### Official Review · Reviewer_1aFR · 2023-07-25

**Soundness:** 2 fair
**Presentation:** 3 good
**Contribution:** 3 good
**Rating:** 7
**Confidence:** 2

**Summary:**

The paper formalizes two variants of the K-FAC optimisation method for a family of neural network architectures that can be expressed as linear layers with weight sharing. Experiments are conducted to validate that the proposed method can be applied to different kinds of neural network architectured speeding-up training.

**Strengths:**

The paper is a bit out of the scope of my specific field of expertise and it is possible that I failed to fully understand all technical details and mathematical formulation, but the general mathematical formulation of the method seems sound and complete. The family or achitectures where the two variants of the proposed method can be applied is formally defined, and the conditions for which exact approximations can be obtained are also defined.

**Weaknesses:**

I miss a better contextualitzation, both from a theoretical and experimental point of view,  of the proposed method with other existing approaches, cited in the related work, that also make use of K-FAC and that have also been applied to similar architectures using convolutional networks, transformers or GNNs. From a theoretical point of view, a better justification of the contribution of the proposed method with respect to other applications of K-FAC. From an experimental point of view, a comparison of the proposed method with other works also using K-FAC.

**Questions:**

See above, in the weaknesses section.  Basically, contextualization of the proposed method with other similar approaches.


-------------------------------


I have read the rebuttal, other reviewers comments, feedback and discussion and I got a better understanding of the contribution of the paper. I increase my rating to accept.

**Limitations:**

Limitations of the method are partially addressed.

---

> ### Author Rebuttal · Authors · 2023-08-09
>
> We thank the reviewer for their time and their feedback.
>
> >I miss a better contextualitzation, both from a theoretical and experimental point of view, of the proposed method with other existing approaches, cited in the related work, that also make use of K-FAC and that have also been applied to similar architectures using convolutional networks, transformers or GNNs. From a theoretical point of view, a better justification of the contribution of the proposed method with respect to other applications of K-FAC. From an experimental point of view, a comparison of the proposed method with other works also using K-FAC.
>
> Thanks for pointing this out. To clarify, K-FAC has only been applied to the architectures we consider here in the form of K-FAC-expand (up to the scaling factor, see the footnote on page 5). While we mention this in lines 50-53 of the paper and in section 3.2, we will emphasise this point more in the next revision of the paper. Since there is no known way to apply K-FAC to the considered architectures besides K-FAC-expand and K-FAC-reduce, we also can’t compare to other variations.
>
> Since you mentioned that the paper is a bit out of the scope of your specific expertise, please don’t hesitate to ask for further clarifications, we understand that it can be hard to place the work into the correct context without familiarity with this specific field of research.

---

> > ### Comment · Reviewer_1aFR · 2023-08-18
> >
> > Dear authors,
> > Than you for your response. After reading all the comments, feedback and discussion I got a better understanding of the contribution of the paper. I will increase my rating to accept.

---

### Official Review · Reviewer_t9fV · 2023-07-27

**Soundness:** 3 good
**Presentation:** 3 good
**Contribution:** 3 good
**Rating:** 6
**Confidence:** 4

**Summary:**

This paper considers linear layers with weight-sharing, which is used in many neural network architectures, such as transformers, convolutional and graph neural networks. Two variants of K-FAC, K-FAC –expand and K-FAC-reduce are considered for two different settings of linear weight sharing layers. The authors also show that they are exact for deep linear networks with weight-sharing in their respective setting. Numerical results show the efficiency of the proposed K-FAC variants.

**Strengths:**

This paper is well-written and organized. The use of second-order methods is typically influenced by the specific network architecture. In this paper, we focus on linear weight-sharing layers, which are crucial and commonly used in Transformer, convolutional, and graph networks, but have not received adequate attention in the realm of second-order methods. In my opinion, a more thoughtful analysis of the architecture could enhance the performance of second-order methods like KFAC and others. Undoubtedly, this paper will expand the applicability of second-order methods and their potential benefits for machine learning tasks.

This paper provides a correct mathematical representation of gradient for the weight-sharing linear layer structure, which is not trivial and can facilitate the development of related optimization methods.

**Weaknesses:**

One of my concerns is the applicability of second-order methods, particularly for modern architectures such as the LLM model. As noted in the AlgoPerf paper[1], the use of KFAC requires detailed architectural information, which may limit its use in practice. In contrast, methods like Shampoo and NG+ appear to offer more flexibility. Additionally, while the direct approach is effective in practice, there is a lack of theoretical understanding regarding the rationale for using the KFAC-reduce approximation.

The computational complexity of the proposed method appears to be higher than previous approaches, which could pose challenges for implementation. However, there is no direct comparison with the current implementation of the KFAC method in this setting, and it would be beneficial to include such a comparison. Additionally, it is recommended to compare the proposed method with other second-order approaches to gauge its effectiveness.

[1] Dahl G E, Schneider F, Nado Z, et al. Benchmarking Neural Network Training Algorithms[J]. arXiv preprint arXiv:2306.07179, 2023.

**Questions:**

To improve the clarity of the paper, it is suggested to provide a detailed description of the current implementations mentioned in section 3.2 [13, 29, 31, 43] in section 2, which will help readers follow the paper more easily and understand the differences more clearly. Besides, some mathematical derivations in appendix can be put in the main text.

It would be better to mention som related works. 1.) the NG+ method [1], which is similar to Shampoo and uses a direct approach to approximate the Fisher matrix.  2.) AdaHessian method [2], which dynamically incorporates the curvature of the loss function via ADAptive estimates of the Hessian matrix.  3.) the SENG method [3], which employs a sketching method to approximate the empirical Fisher matrix and is implemented in the ASDL package.

The computational complexity of using regular K-FAC approximation over N R terms instead of the usual N terms in the KFAC-expand method is higher, and when the batch size is large, as reported in the numerical experiments, it also increases memory usage.

In the KFAC-reduce setting, where $\widehat{A_l}$ (also $\widehat{B_l}$) is a symmetric matrix with rank at most N, this rank is small when a smaller batch size is used. However, it is unclear whether this reduction preserves sufficient curvature information. Do the authors consider use moving averaging in the implementation?


[1] Yang, Minghan, et al. "An efficient fisher matrix approximation method for large-scale neural network optimization." IEEE Transactions on Pattern Analysis and Machine Intelligence 45.5 (2022): 5391-5403.

[2] Yao, Zhewei, et al. "Adahessian: An adaptive second order optimizer for machine learning." proceedings of the AAAI conference on artificial intelligence. Vol. 35. No. 12. 2021.

[3] Yang, Minghan, et al. "Sketch-based empirical natural gradient methods for deep learning." Journal of Scientific Computing 92.3 (2022): 94.

---

> ### Author Rebuttal · Authors · 2023-08-09
>
> We thank the reviewer for their time and the detailed comments.
>
> >One of my concerns is the applicability of second-order methods, particularly for modern architectures such as the LLM model. As noted in the AlgoPerf paper [1], the use of KFAC requires detailed architectural information, which may limit its use in practice. (...)
>
> LLMs specifically can be easily implemented in a way that is compatible with K-FAC (in fact, many popular implementations already are, e.g. nanoGPT, which implements GPT-2).
> To demonstrate this, **we compute a K-FAC GGN approximation for nanoGPT on the full DART dataset** [1]. In line with the complexities and our other results K-FAC-reduce is significantly faster, only taking 70% of the time of K-FAC-expand. Please find the table with the results in our reply to Reviewer hvpV.
>
> More generally, we agree that second-order methods, and K-FAC in particular, are often not easy to apply to modern neural network architectures. However, we think that our work contributes to extending its applicability, since we describe how K-FAC can be applied to basically arbitrary neural network layers (as long as they can be expressed as linear weight-sharing layers). Despite this theoretical extension, we fully agree that K-FAC is still not agnostic to the model implementation, whereas other methods like Shampoo are. Since K-FAC has a more direct connection to the Fisher/GGN than the preconditioner used in Shampoo and other similar methods, we believe it is still a valuable tool to study the behaviour of second-order methods in deep learning. But, as mentioned before, it might also be able to significantly speed up training in specific use cases like LLM training.
>
> >Additionally, while the direct approach is effective in practice, there is a lack of theoretical understanding regarding the rationale for using the KFAC-reduce approximation.
>
> K-FAC-reduce is just as well motivated as K-FAC-expand, as they are derived by following the same principles, but in two different settings.
>
> >The computational complexity of the proposed method appears to be higher than previous approaches, which could pose challenges for implementation. However, there is no direct comparison with the current implementation of the KFAC method in this setting, and it would be beneficial to include such a comparison.
>
> There seems to be a misunderstanding: K-FAC-reduce has actually lower computational complexity than K-FAC-expand, which has previously been used for linear weight-sharing layers. There is no third way of applying K-FAC to linear weight-sharing layers in the literature that we are aware of.
>
> >Additionally, it is recommended to compare the proposed method with other second-order approaches to gauge its effectiveness.
>
> While we fully agree that it will be interesting to see how K-FAC compares to other second-order approaches, we decided to not focus on this question in our paper. We do not intend to make any general claims about K-FAC as an optimiser, but want to generalise K-FAC to modern neural network architectures.
>
> To make more general statements about K-FAC’s performance as a general purpose optimiser we would have to use a different experimental setup, e.g. run a valid submission to the AlgoPerf benchmark.
>
> >To improve the clarity of the paper, it is suggested to provide a detailed description of the current implementations mentioned in section 3.2 [13, 29, 31, 43] in section 2, which will help readers follow the paper more easily and understand the differences more clearly. (...)
>
> We will highlight the differences between the different implementations more clearly and will use the additional page to move some parts of the derivations to the main text. To clarify, K-FAC has only been applied to the architectures we consider here in the form of K-FAC-expand (up to the scaling factor, see the footnote on page 5). While we mention this in lines 50-53 of the paper and in section 3.2, we will highlight this point in the revision of the paper. Also, note that we provide a Python code snippet in Listing 1 in the appendix, illustrating the actual implementation.
>
> >It would be better to mention some related works. 1.) the NG+ method [1], which is similar to Shampoo and uses a direct approach to approximate the Fisher matrix. 2.) AdaHessian method [2], which dynamically incorporates the curvature of the loss function via ADAptive estimates of the Hessian matrix. 3.) the SENG method [3], which employs a sketching method to approximate the empirical Fisher matrix and is implemented in the ASDL package.
>
> Thank you for the suggestions, we will include these references in section 2.1.
>
> >The computational complexity of using regular K-FAC approximation over N R terms instead of the usual N terms in the KFAC-expand method is higher, and when the batch size is large, as reported in the numerical experiments, it also increases memory usage.
>
> Yes, this is correct. In fact, K-FAC-expand is the currently used K-FAC variation for linear weight-sharing layers and the here derived K-FAC-reduce offers an alternative with lower computational complexity by a factor of $R$.
>
> >In the KFAC-reduce setting, where (also ) is a symmetric matrix with rank at most N, this rank is small when a smaller batch size is used. However, it is unclear whether this reduction preserves sufficient curvature information. Do the authors consider use moving averaging in the implementation?
>
> We agree that it will be interesting to see how moving averages over the Kronecker factors will influence the approximation quality and downstream optimisation performance, but only use local estimates in our experiments, which seems to be enough to achieve the reported performance.
>
> We hope that we have sufficiently addressed your concerns. Please let us know if there still is anything unclear.
>
> [1] L. Nan et al. DART: Open-Domain Structured Data Record to Text Generation. NAACL-HLT, 2021.

---

> > ### Comment · Reviewer_t9fV · 2023-08-18
> > **Reply to Rebuttal by Authors**
> >
> > I apologize for the delayed response to the rebuttal. Thank you for addressing my questions, as it has helped me gain a clearer understanding of this paper.
> >
> > Training large transformer models still poses a challenge for second-order methods. The nanoGPT experiments are encouraging, although the number of parameters in nanoGPT (as indicated in https://github.com/karpathy/nanoGPT) is still smaller compared to the commonly used Transformer models such as GPT-3 and GPT-4. It would be beneficial if the authors reported the train/val loss and compared KFAC with other methods like AdamW.
> >
> > I understand that the authors' focus is on generalizing K-FAC to modern neural network architectures. However, not everyone in the deep learning community is aware of the effectiveness of KFAC. To enhance the persuasiveness of the paper, it would be better to include other first and second-order methods in the numerical experiments.
> >
> > Furthermore, I still have some questions listed below.
> >
> > The method K-FAC is specifically developed to approximate the Fisher Information Matrix (FIM). In general cases, the Generalized Gauss-Newton (GGN) is not equivalent to the FIM. The reference [7] mentioned in section 2.4 can be viewed as a Kronecker-factored approximation to GGN, rather than "KFAC to approximate GGN". I understand that the approach to approximate Fisher can be extended to the GGN case. However, these two concepts should be distinguished.
> >
> > There appears to be a mistake in equation 26. Could you please explain how the second equation was derived? In general cases,
> > $(A \otimes B) D (A^\top \otimes B^\top) \neq (AA^\top) \otimes (BDB^\top)$, where $A\in R^{p\times 1}$, $B\in R^{b\times C}$, $D\in R^{C\times C}$, and $C > 1$. Since Lemma 3 relies on equation 26, please provide clarification on this matter.
> >
> > The case in KFAC-expand and the case in [13] are different. In [13], the authors do not consider the loss with N·R terms. Why does line 199 state that the approximation (6) has been derived for CNN? In the authors' previous reply, the statement "K-FAC-expand is the currently used K-FAC variation for linear weight-sharing layers" is not applicable to the CNN case.
> >
> > I noticed the idea in ``Eva: A General Vectorized Approximation Framework for Second-order Optimizationv'' is very similar to the KFAC-reduce. Please see eq 9 in that paper.

---

> > > ### Author Response · Authors · 2023-08-18
> > >
> > > > Training large transformer models still poses a challenge for second-order methods. (...)
> > >
> > > We fully agree that training large models still poses challenges for second-order methods, probably mostly due to the memory overhead. We believe this could be addressed by considering sparse structures to approximate the Kronecker factors, see e.g. [1]. However, this is beyond the scope of our paper.
> > > While we could not provide full training results for GPT-2 due to the time constraints of the rebuttal, we agree that this will be an interesting experiment.
> > >
> > > > I understand that the authors' focus is on generalizing K-FAC to modern neural network architectures. (...)
> > >
> > > The first-order baseline we show per experiment is the best one among AdamW, NAdamW, SGD with Nesterov momentum, and SGD with heavy ball momentum, as described in appendix C.1.
> > > Comparing the K-FAC variations to other second-order methods is definitely interesting and could increase the appeal to some readers, but we do not think that this is necessary for the central claims and contributions of the paper. However, we are happy to add at least one second-order baseline for the camera-ready version, e.g. Shampoo.
> > >
> > >
> > > > The method K-FAC is specifically developed to approximate the Fisher Information Matrix (FIM). In general cases, the Generalized Gauss-Newton (GGN) is not equivalent to the FIM. (...)
> > >
> > > We agree that the two approximations should be distinguished in the case where the FIM and the GGN do not coincide. For the cases where they do, which include common loss functions like the cross-entropy and mean squared error loss we consider in the paper, the two approximations are exactly identical. Hence why we decided to also talk about "K-FAC to approximate the GGN". We will add a clarifying sentence to the paper.
> > >
> > >
> > > > There appears to be a mistake in equation 26. (...) In general cases, $(A \\otimes B) D (A^T \\otimes B^T) \\neq (AA^T) \\otimes (BDB^T)$, where $A \\in \\mathbb{R}^{p \\times 1}$, $B \\in \\mathbb{R}^{b \\times C}$, $D \\in \\mathbb{R}^{C \\times C}$, and $C > 1$. Since Lemma 3 relies on equation 26, please provide clarification on this matter.
> > >
> > > Sure. The crucial detail is that $A \\in \\mathbb{R}^{p \\times 1}$ is a vector, as you correctly define. In this case the equality indeed holds. To see this, we can rephrase your claim to $(A \\otimes B) D \\neq (A \\otimes BD),$ since if this was an equality we could write $(A \\otimes BD) (A^T \\otimes B^T) = (AA^T) \\otimes (BDB^T)$ due to the mixed-product property of the Kronecker product. We will now show that this inequality is in fact an equality. We have
> > > $$(A \\otimes B) D = \\begin{bmatrix} A_1B \\ ... \\ A_pB \\end{bmatrix}^T D = \\begin{bmatrix} A_1BD \\ ... \\ A_pBD \\end{bmatrix}^T = (A \\otimes BD),$$
> > > where $A_1, ..., A_p$ are the scalar components of the vector $A$.
> > >
> > > > The case in KFAC-expand and the case in [13] are different. In [13], the authors do not consider the loss with N·R terms. (...)
> > >
> > > In [13] the authors derive K-FAC-expand for CNNs by simply explicitly stating the independence assumptions it implies and not by arguing about the exactness in the deep linear case.
> > >
> > > We have to distinguish between the setting of K-FAC-expand, i.e. a loss with NR terms, and the approximation itself. As explained in the paper, we can also use K-FAC-expand in the reduce setting and K-FAC-reduce in the expand setting -- which is also what we do in the experiments, since we apply both approximations in the same setting. Crucially, K-FAC-expand was the only approximation known for both settings. However in the reduce setting, K-FAC-expand cannot be motivated by making statements about the exactness in the deep linear case with a Gaussian likelihood. In contrast, we can motivate K-FAC-reduce in this way in the case of a CNN for image classification (assuming average pooling as the aggregation function). We state this in lines 240-242 of the paper, below Proposition 2.
> > >
> > > To provide some intuition on why K-FAC-expand and K-FAC-reduce can both always be used for linear weight-sharing layers, please refer to the Python code snippet in Listing 1 in the appendix. Here you can see that the quantities needed to calculate the K-FAC approximation will have the same shape in both settings, allowing us to arbitrarily choose one of the two K-FAC variations to deal with the weight-sharing dimension.
> > >
> > > > I noticed the idea in "Eva(...)" is very similar to the KFAC-reduce. (...)
> > >
> > > Thanks for the reference. The difference is that the approximation in Eva sums over the mini-batch dimension, whereas K-FAC-reduce takes a sum/mean over the weight-sharing dimension. This is quite different, since Eva does not have a clear motivation in terms of the connection to the exact Fisher/GGN, leads to rank-1 Kronecker factors, and is completely unrelated to weight-sharing.
> > >
> > > [1] R. Grosse, J. Bae, C. Anil et al. Studying Large Language Model Generalization with Influence Functions. Preprint (arXiv: 2308.03296), 2023.

---

> > > > ### Comment · Reviewer_t9fV · 2023-08-18
> > > > **Reply**
> > > >
> > > > Thanks for you reply. All my questions are addressed. I would increase my score from 5 to 6.

---

### Decision · Program_Chairs · 2023-09-21

**Decision:**

Accept (spotlight)

**Comment:**

The paper generalizes the second-order KFAC optimization method to linear weight-sharing layers. The authors sufficiently addressed the reviewers' concerns regarding the applicability of the method to large LLMs and comparison to other second-order methods. The technique is useful for making second-order methods applicable to large networks and encourages further research in this important area.